# ALIGNDRIVE: ALIGNED LATERAL-LONGITUDINAL PLANNING FOR END-TO-END AUTONOMOUS DRIVING

## ABSTRACT

End-to-end autonomous driving has rapidly progressed, enabling joint perception and planning in complex environments. In the planning stage, state-of-the-art (SOTA) end-to-end autonomous driving models decouple planning into parallel lateral and longitudinal predictions. While effective, this parallel design can lead to i) coordination failures between the planned path and speed, and ii) underutilization of the drive path as a prior for longitudinal planning, thus redundantly encoding static information. To address this, we propose a novel cascaded framework that explicitly conditions longitudinal planning on the drive path, enabling coordinated and collision-aware lateral and longitudinal planning. Specifically, we introduce a path-conditioned formulation that explicitly incorporates the drive path into longitudinal planning. Building on this, the model predicts longitudinal displacements along the drive path rather than full 2D trajectory waypoints. This design simplifies longitudinal reasoning and more tightly couples it with lateral planning. Additionally, we introduce a planning-oriented data augmentation strategy that simulates rare safety-critical events, such as vehicle cut-ins, by adding agents and relabeling longitudinal targets to avoid collision. Evaluated on the challenging Bench2Drive benchmark, our method sets a new SOTA, achieving a driving score of 89.07 and a success rate of 73.18%, demonstrating significantly improved coordination and safety. Visualizations are provided at this webpage.[1].

## 1 INTRODUCTION

End-to-end (E2E) autonomous driving has made rapid progress in recent years, achieving increasingly sophisticated perception-planning capabilities (Sun et al., 2024; Guo et al., 2025; Jia et al., 2025; Song et al., 2025). Since UniAD (Hu et al., 2023), end-to-end approaches have commonly relied on explicit BEV feature maps in conjunction with query-based architectures to bridge perception and planning (Hu et al., 2023; Weng et al., 2024). More recent works have started to bypass BEV features, directly mapping sensor inputs to planned trajectories or intermediate latent representations (Jia et al., 2025; Sun et al., 2024). Within this line, several studies have shown that disentangling lateral and longitudinal planning at the planning stage can be particularly beneficial (Jaeger et al., 2023; Renz et al., 2024). In this paradigm, lateral planning predicts the drive path—waypoints sampled at fixed spatial intervals—as the target for steering, while longitudinal planning predicts the trajectory—waypoints sampled at fixed temporal intervals—as the target for speed control.

Among these, one of the most recent works, HiP-AD (Tang et al., 2025) achieves multi-modal prediction by initializing multiple paired drive path and trajectory queries, with each query in the pair decoded by an independent head. While this design delivers strong results, we argue that planning the drive path and trajectory through two independent branches introduces two key drawbacks: (i) splitting planning into two independent branches makes it difficult to enforce kinematic consistency between the outputs. For example, as shown in the top-right of Fig 1(b), a lateral path requiring a sharp turn and a longitudinal trajectory demanding high speed are not constrained to be mutually consistent during training, potentially leading to inconsistent predictions that challenge downstream execution. This happens because the longitudinal branch does not explicitly leverage the drive path as a prior, leading to misaligned lateral and longitudinal decisions; (ii) the trajectory prediction relies on static scene elements such as road geometry and lane structure, which are already captured by the drive path. Re-encoding these cues in the longitudinal branch is redundant and limits the model's focus on dynamic interactions, which are critical for safe and effective longitudinal planning.

---

[1] A copy of videos has been included in the supplementary materials

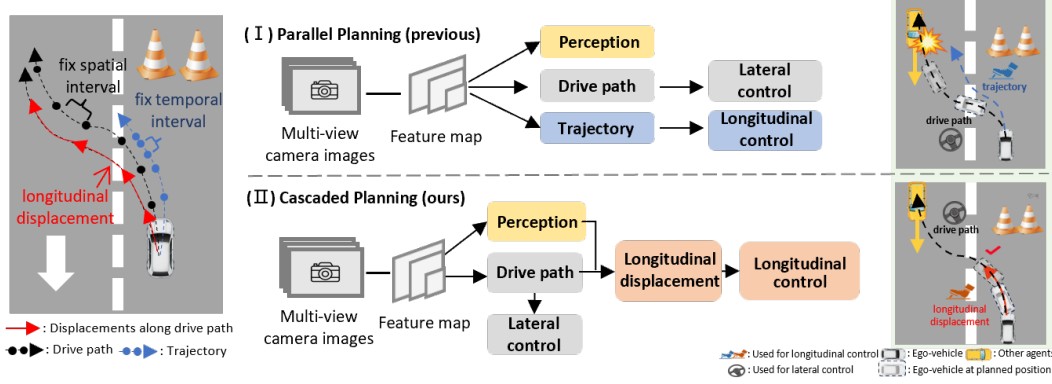

**(a) Planning Targets**      **(b) E2E Paradigms**

Figure 1: (a) Drive path (black), trajectory (blue), and longitudinal displacement (red). Path waypoints are sampled spatially, trajectory waypoints temporally, and displacements represent traveled distance along the path at fixed time intervals. (b) Comparison of E2E paradigms. Parallel planning predicts the drive path and longitudinal trajectory independently, which can lead to potential coordination inconsistencies. In the example on the right, the independently predicted longitudinal trajectory is collision-free by itself, but applying its speed along a separately predicted lateral path could cause a collision. In contrast, our cascaded paradigm first predicts the drive path and then regresses path-conditioned longitudinal displacements. With the path prior, the model identifies the potential conflict and outputs shorter displacements, yielding to avoid collision. Perception inputs are omitted for clarity.

To address these limitations, we propose a new cascaded paradigm that establishes a tight coupling between lateral and longitudinal planning via an anchor-based regression framework, where longitudinal planning is conditioned on the lateral drive path. Building on this foundation, we further simplify the task by predicting longitudinal displacements along the drive path instead of full 2D trajectory waypoints, while still providing effective targets for longitudinal control, as illustrated in Fig. 1(a). This formulation naturally couples lateral and longitudinal planning: the drive path provides a stable geometric prior, while longitudinal reasoning is simplified to predicting displacement conditioned on dynamic agents. By decoupling lateral geometry from longitudinal reasoning, the framework allows the model to focus on dynamic interactions and improves collision-aware planning, as shown in Fig. 1(b).

This cascaded formulation also unlocks a highly effective, planning-oriented data augmentation strategy. Since our longitudinal plan consists of simple displacement values along a fixed path, we can realistically simulate safety-critical events like vehicle cut-ins—which are rare in real-world logs—by programmatically shortening the displacement distances in response to inserted agents, without altering the lateral path. This targeted data augmentation exposes the planner to a rich set of critical scenarios, substantially improving its collision avoidance capabilities.

Building on these insights, we develop an E2E driving framework AlignDrive that conditions longitudinal planning on the drive path and leverages this formulation to enable effective data augmentation, with code and models to be publicly released. Overall, our contributions are threefold:

- We propose a novel cascaded planning paradigm where longitudinal planning is explicitly conditioned on a predicted lateral drive path. This method establishes a tight coupling between the two tasks, using the path as geometric priors for subsequent longitudinal planning.

- Based on this paradigm, we reformulate the longitudinal planning task as a simpler 1D displacement prediction problem along the drive path. This allows the model to focus its capacity on crucial dynamic interactions rather than redundantly encoding static geometry.

- We introduce an effective, planning-oriented data augmentation strategy. By programmatically modifying only the 1D displacement labels in response to inserted agents, we can generate diverse and realistic safety-critical training scenarios that are rare in logged data.

Our experiments conducted on the popular closed-loop simulator benchmark Bench2Drive (Jia et al., 2024) demonstrate that AlignDrive outperforms state-of-the-art driving techniques.

## 2 RELATED WORK

### 2.1 END-TO-END AUTONOMOUS DRIVING

End-to-end autonomous driving methods (Wang et al., 2025; Tang et al., 2025; Gao et al., 2025; Sun et al., 2024; Hu et al., 2023; Jiang et al., 2023; Xing et al., 2025) have rapidly advanced in recent years, with trajectory planning playing a central role in predicting the ego vehicle's future states. One line of work, exemplified by SparseDrive (Sun et al., 2024; Song et al., 2025), directly predicts trajectories in an end-to-end manner. While effective in nominal scenarios, this joint prediction paradigm often struggles to achieve accurate lateral and longitudinal plannning simultaneously. TF++ (Jaeger et al., 2023) predicts the drive path and instantaneous vehicle speed in parallel, with speed treated as a classification task. However, the coarse discretization of velocity limits planning accuracy. More recent approaches, including HiP-AD (Tang et al., 2025) and Carllava (Renz et al., 2024), instead decouple path and trajectory prediction. HiP-AD employs independent heads, while Carllava, built on a LLaVA-like architecture (Liu et al., 2023), generates the drive path and trajectory sequentially as output tokens. However, these methods still rely on predicting full waypoints rather than explicit longitudinal displacements, limiting precise alignment between lateral and longitudinal planning in challenging scenarios such as sharp turns or dynamic interactions. In contrast, we propose a cascaded, anchor-based formulation that first predicts the drive path and then forecasts a sequence of future longitudinal displacements along it. Our approach regresses offsets from predefined anchors using a two-stage design with dedicated modules for path and displacement prediction, rather than jointly generating them as tokens. This naturally enforces lateral–longitudinal consistency, simplifies reasoning about dynamic interactions, and improves path-following safety.

### 2.2 DATA AUGMENTATION

Data augmentation is widely employed in multiple fields (Zhang et al., 2022; Qiu et al., 2025; Wu et al., 2023; Lin et al., 2022). In autonomous driving, it is commonly applied to augment image data through techniques such as cropping, flipping, and color jittering (Sun et al., 2024), which improve the model's ability to generalize across varying visual conditions and strengthen perception robustness. Pluto (Cheng et al., 2024) employs agent drop and insertion as data augmentation strategies to generate both positive and negative scene samples. These augmented samples are utilized in a contrastive learning framework to enhance the model's scene representation capabilities. However, these augmentations primarily affect perception and influence planning only indirectly. TF++ (Jaeger et al., 2023) introduced an auxiliary camera in the simulation environment, which is randomly repositioned at each time step to increase data diversity. This approach relies on additional simulator equipment and focuses primarily on lateral recovery, providing limited guidance for longitudinal planning or dynamic interaction reasoning. In contrast, our planning-oriented augmentation is directly coupled with longitudinal planning, operating on the percepted agents and adjusting longitudinal displacements. This forces the model to focus explicitly on dynamic agent interactions, enabling path-consistent and collision-aware planning in rare, safety-critical scenarios.

## 3 METHOD

### 3.1 OVERVIEW

Figure 2 provides an overview of AlignDrive, which consists of three main components. The **Drive Path Predictor** refines queries via cross-attention with image features (Lin et al., 2022), producing representations of the drive path, map, and dynamic agents. The **Planning-oriented Data Augmentation** module decodes agent queries into bounding boxes, re-encodes them as structured features, and enables insertion of synthetic agents with relabeled longitudinal displacements for consistent supervision. The **Longitudinal Planning** module then predicts displacements along the drive path from enriched queries, ensuring spatial consistency and collision awareness. This design preserves end-to-end training while supporting robust planning. We discuss components below.

### 3.2 DRIVE PATH PREDICTOR

Let us denote multi-scale features from $V$ camera views as $\{\mathbf{f}_i\}_{i=1}^V$. Based on training data, we cluster ground-truth annotations to obtain anchors, which differ by task modality: bounding boxes for agents, and typical polylines for map elements and drive paths. We denote them as $\mathbf{A}_a \in \mathbb{R}^{N_a \times D_a}$, $\mathbf{A}_m \in \mathbb{R}^{N_m \times D_m}$, and $\mathbf{A}_d \in \mathbb{R}^{N_d \times D_d}$, where $D_a, D_m, D_d$ are the dimensions of each type. Based on these anchors, we initialize three sets of task queries: agent queries $\mathbf{Q}_a \in \mathbb{R}^{N_a \times C}$, map queries $\mathbf{Q}_m \in \mathbb{R}^{N_m \times C}$, and drive path queries $\mathbf{Q}_d \in \mathbb{R}^{N_d \times C}$, where $C$ is the feature dimension. The

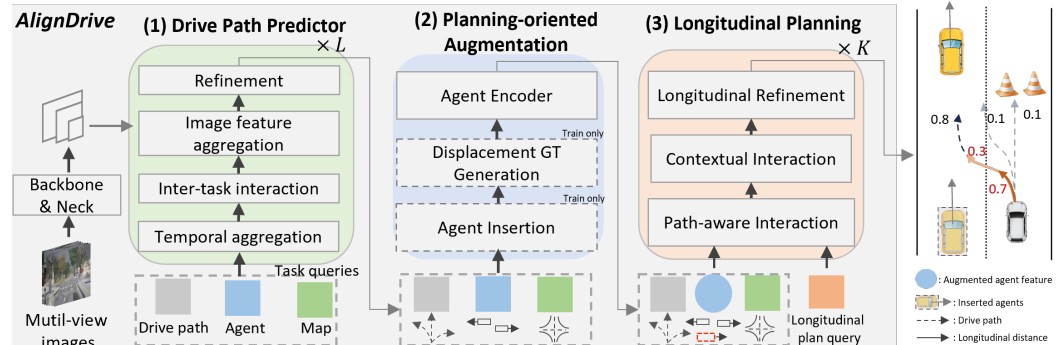

Figure 2: Overview of the proposed AlignDrive system, which consists of three components. The Drive Path Predictor refines queries through cross-attention with image features to encode the drive path, maps, and agents. The Planning-oriented Data Augmentation enriches scenarios by inserting additional agents and relabeling longitudinal displacements. Finally, the Longitudinal Planning Module predicts forward displacements along the drive path; combined with the path, these displacements yield the final trajectory that is both collision-aware and spatially consistent. On the right side of the figure, the black numbers denote the scores of predicted drive paths, while the red numbers represent the scores of the corresponding longitudinal planning for each drive path.

Drive Path Predictor consists of $L$ stacked blocks, within which queries interact with image features, historical information, and each other. Through these interactions, the corresponding anchors are iteratively refined across blocks, yielding progressively updated estimates.

**Temporal Aggregation.** To incorporate historical information, each query interacts with retained queries from previous frames via a top-$k$ strategy:

$$\mathbf{Q}_d \leftarrow \text{Cross-attention}\left(\mathbf{Q} = \mathbf{Q}_d, \mathbf{K} = \mathbf{Q}_d^{t-T_p:t-1}, \mathbf{V} = \mathbf{Q}_d^{t-T_p:t-1}\right), \quad (1)$$

where $\mathbf{Q}_d$ is the current drive path queries, and $\mathbf{Q}_d^{t-T_p:t-1}$ are historical ones. Map queries $\mathbf{Q}_m$ and agent queries $\mathbf{Q}_a$ are updated similarly.

**Inter-Task Interaction.** We enable interactions among drive path, agent, and map queries through cross-attention, allowing path queries to be contextually aware of agents and maps, and constraining agent behaviors with map information.

$$\mathbf{Q}_d \leftarrow \text{Cross-attention}\left(\mathbf{Q} = \mathbf{Q}_d, \mathbf{K} = [\mathbf{Q}_a \| \mathbf{Q}_m], \mathbf{V} = [\mathbf{Q}_a \| \mathbf{Q}_m]\right),$$
$$\mathbf{Q}_a \leftarrow \text{Cross-attention}\left(\mathbf{Q} = \mathbf{Q}_a, \mathbf{K} = \mathbf{Q}_m, \mathbf{V} = \mathbf{Q}_m\right), \quad (2)$$

where $[\cdot \| \cdot]$ denotes concatenation along the token dimension.

**Image feature aggregation.** To fuse image features, anchors are projected onto multi-view images, and their sampled features are aggregated via deformable attention. For drive path queries:

$$\mathbf{Q}_d \leftarrow \text{DA}\left(\mathbf{Q} = \mathbf{Q}_d, \mathbf{K} = \mathcal{P}(\mathbf{A}_d, \{\mathbf{F}_i\}_{i=1}^V), \mathbf{V} = \mathcal{P}(\mathbf{A}_d, \{\mathbf{F}_i\}_{i=1}^V)\right), \quad (3)$$

where DA is deformable attention and $\mathcal{P}(\cdot)$ denotes projection and sampling. Map queries $\mathbf{Q}_m$ and agent queries $\mathbf{Q}_a$ are enhanced in the same way using their anchors $\mathbf{A}_m, \mathbf{A}_a$.

**Refinement.** The model iteratively refines its predictions across the L blocks. In the refinement stage of each block, for a given anchor $\mathbf{A}_d$, we first generate a feature embedding using a task-specific encoder, $MLP_{enc}(\mathbf{A}_d)$. This embedding is fused with the corresponding query $\mathbf{Q}_d$ and fed into an MLP to predict a corrective offset, $\Delta \mathbf{Y}_d$. The anchor is then updated by applying this offset. This process allows the model to progressively improve its estimate from a coarse anchor to a precise prediction.

$$\Delta \mathbf{Y}_d = \text{MLP}\left(\mathbf{Q}_d + MLP_{enc}(\mathbf{A}_d)\right), \quad \mathbf{A}_d \leftarrow \mathbf{A}_d + \Delta \mathbf{Y}_d, \quad (4)$$

where $\Delta \mathbf{Y}_d \in \mathbb{R}^{N_d \times D_d}$ is the predicted offset for each drive path anchor, and $D_d = P \times 2$ corresponds to $P$ future waypoints. The refined waypoints are obtained as $\hat{\mathbf{Y}}_d = \mathbf{A}_d$ after iteratively updating the anchors through all $L$ blocks. A separate MLP head is applied to the drive path query

(a) Planning-oriented augmentation patterns (b) Representation encoding

Figure 3: (a) Planning-oriented augmentation. Non-threatening agents are inserted at a distance with unchanged GT displacements, while threatening agents are placed nearby and cause adaptive shortening of GT displacements. (b) Representation encoding. Inserted agents are projected into future positions, transformed to corner representations, and encoded via a Fourier encoder (top). Reference points are sampled by displacement anchors and encoded with MLPs. For clarity, although multiple drive paths are predicted in practice, only one representative path is illustrated here. (bottom)

to predict confidence scores $\mathbf{S}_d \in \mathbb{R}^{N_d \times 1}$ for candidate drive paths. Map anchors are refined in the same iterative manner using $\mathbf{Q}_m$ and $\mathbf{A}_m$.

For agents, static attributes (e.g., position, size, orientation) are predicted using an MLP applied to the queries combined with anchor features, while dynamic motion is predicted directly from queries without anchor-based refinement:

$$\hat{\mathbf{Y}}_a^{state} = \text{MLP}_{state}\left(\mathbf{Q}_a + E(\mathbf{A}_a)\right), \quad \hat{\mathbf{Y}}_a^{motion} = \text{MLP}_{motion}\left(\mathbf{Q}_a\right), \tag{5}$$

where $\hat{\mathbf{Y}}_a^{state} \in \mathbb{R}^{N_a \times S}$ contains $S$ attributes for each agent, and $\hat{\mathbf{Y}}_a^{motion} \in \mathbb{R}^{N_a \times T \times 2}$ contains the predicted future trajectories.

### 3.3 PLANNING-ORIENTED DATA AUGMENTATION

To enrich interactive scenarios, we insert a virtual agent into the detected agents with probability $\alpha$.

**Agent Insertion.** The virtual agent is initialized with a randomly sampled state $\mathbf{X}_{vir}$ and a target point $\mathbf{P}^*$, selected on the ego vehicle's ground-truth drive path $\mathbf{Y}_d^{\text{GT}}$. Together, they determine the virtual agent's future motion. As illustrated in Fig. 3(a), the virtual agent randomly adopts one of two velocity patterns: gradually approaching from afar, making it a low-risk and safe maneuver, or approaching faster and closer, potentially colliding with the ego vehicle. To maintain the total number of agents, the original agent with the lowest confidence is removed, and the virtual agent is inserted in its place. The resulting augmented set of agent states and motions are denoted as $\tilde{\mathbf{Y}}^{state}$ and $\tilde{\mathbf{Y}}^{motion}$, ensuring exposure to challenging interactions while preserving scene consistency.

**Displacement Ground-truth Generation.** The ground-truth displacement label is defined as a sequence $\mathbf{Y}_d^{\text{GT}} \in \mathbb{R}^{T \times 1}$, representing $T$ future longitudinal displacements of the ego vehicle along the drive path, each measured over a fixed temporal interval. When a virtual agent is inserted that would collide with the ego vehicle, we first determine the maximum total displacement $D_{\text{safe}}$ the ego vehicle can travel over the $T$ steps without causing a collision. Given the original total displacement $D_{\text{orig}} = \sum_{t=1}^T \mathbf{Y}_d^{\text{GT}}[t]$, we compute a safe scaling factor $\beta = D_{\text{safe}}/D_{\text{orig}}$ and uniformly scale the original sequence: $\tilde{\mathbf{Y}}_d^{\text{GT}} = \beta \cdot \mathbf{Y}_d^{\text{GT}}$. This procedure ensures that the displacement at each step is consistently reduced to avoid collisions, while maintaining a dynamically plausible motion profile.

**Agent Encode.** After augmentation, agents are represented using both their states and motions, denoted as $\tilde{\mathbf{Y}}^{state}$ and $\tilde{\mathbf{Y}}^{motion}$. These variables are converted into a unified corner-based representation:

$$\mathbf{C}_a = f_{\text{corner}}(\tilde{\mathbf{Y}}^{state}, \tilde{\mathbf{Y}}^{motion}), \quad \mathbf{C}_a \in \mathbb{R}^{N_a \times (T+1) \times 8}, \tag{6}$$

where $f_{\text{corner}}(\cdot)$ converts each agent's bounding box state and future trajectory into the coordinates of its four corners across all time steps, as illustrated in the top of Fig. 3(b). To construct agent features, we separately process geometry and category information. The corner representation $\mathbf{C}_a$ is first mapped using Fourier encoder $\Phi$ and passed through $\text{MLP}_{\text{corner}}$ (Mildenhall et al., 2021), while the agent category (extracted from $\tilde{\mathbf{Y}}^{state}$) is mapped with $\Phi$ and processed through $\text{MLP}_{\text{category}}$. Their outputs are then summed to form the unified agent embedding:

$$\mathbf{E}_a^{0:T} = \text{MLP}_{\text{corner}}(\Phi(\mathbf{C}_a)) + \text{MLP}_{\text{category}}(\Phi(\text{category})), \quad \mathbf{E}_a^{0:T} \in \mathbb{R}^{N_a \times (T+1) \times C}. \tag{7}$$

Here, $\mathbf{E}_a^t \in \mathbb{R}^{N_a \times C}$ denotes the features of all agents at time step $t$, providing a temporally consistent, category-aware representation. The detailed process can be found in the appendix.

## 3.4 LONGITUDINAL PLANNING MODULE

The longitudinal planning module predicts a sequence of ego vehicle's future displacements along the drive path at fixed time intervals. It is implemented as $K$ stacked blocks, where queries interact with agent and path features and are progressively refined. We formulate this as an *anchor-based offset regression* task: for each candidate path, $M$ anchors are defined, each representing a sequence of longitudinal displacements for the current step and $T$ future steps, yielding $M \times (T+1)$ learnable queries. These queries are responsible for predicting offsets relative to their anchors, thereby coupling the drive path geometry with agent interactions and enabling precise longitudinal planning.

$$\mathbf{A}_l^{0:T} \in \mathbb{R}^{N_d \times M \times (T+1) \times 1}, \quad \mathbf{Q}_l^{0:T} \in \mathbb{R}^{N_d \times M \times (T+1) \times C}, \tag{8}$$

where the superscript $0:T$ denotes the stacked sequence of time steps, covering from the current step ($t = 0$) to $T$ future steps. Here, $\mathbf{A}_l^{0:T}$ are the anchor displacements and $\mathbf{Q}_l^{0:T}$ their learnable queries. The final displacements are obtained by adding predicted offsets to the anchors.

**Path-aware Interaction.** Each longitudinal planning query is enhanced with reference waypoints sampled along the predicted drive path $\hat{\mathbf{Y}}_d$ at anchor displacements $\mathbf{A}_l^{0:T}$. Let sampled points $\mathbf{P}_l$ be

$$\mathbf{P}_l = \text{Interp}(\hat{\mathbf{Y}}_d, \mathbf{A}_l^{0:T}), \quad \mathbf{P}_l \in \mathbb{R}^{N_d \times M \times (T+1) \times 2}, \tag{9}$$

where $\text{Interp}(\cdot)$ denotes linear interpolation along the path according to cumulative anchor displacements. The points for each anchor are encoded jointly into a feature vector, as shown in Fig. 3(b) (bottom):

$$\mathbf{F}_l = \text{MLP}(\mathbf{P}_l), \quad \mathbf{F}_l \in \mathbb{R}^{N_d \times M \times C}. \tag{10}$$

Then, this feature is broadcasted across the T+1 time steps and added to the longitudinal query for cross-attention with the drive path queries $\mathbf{Q}_d$:

$$\mathbf{Q}_l^{0:T} \leftarrow \text{CrossAttn}(\mathbf{Q} = \mathbf{Q}_l^{0:T} + \mathbf{F}_l, \mathbf{K} = \mathbf{Q}_d, \mathbf{V} = \mathbf{Q}_d). \tag{11}$$

In this way, each longitudinal planning query incorporates both the geometry of the drive path and its anchor-based temporal reference.

**Contextual Interaction.** At each time step $t$, longitudinal planning queries interact with dynamic agents via cross-attention on the encoded agent features to capture agent-specific context:

$$\mathbf{Q}_l^t \leftarrow \text{CrossAttn}(\mathbf{Q} = \mathbf{Q}_l^t, \mathbf{K} = \mathbf{E}_a^t, \mathbf{V} = \mathbf{E}_a^t), \quad t = 0, \ldots, T. \tag{12}$$

The updated queries attend to map queries via cross-attention, incorporating static cues such as stop lines for longitudinal planning:

$$\mathbf{Q}_l^{0:T} \leftarrow \text{CrossAttn}(\mathbf{Q} = \mathbf{Q}_l^{0:T}, \mathbf{K} = \mathbf{Q}_m, \mathbf{V} = \mathbf{Q}_m). \tag{13}$$

Finally, a temporal positional encoding is added to the queries, and causal self-attention is applied along the temporal dimension to enforce consistency:

$$\mathbf{Q}_l^{0:T} \leftarrow \text{CausalSelfAttn}(\mathbf{Q}_l^{0:T} + \text{PE}^{0:T}). \tag{14}$$

**Longitudinal Refinement.** After obtaining the updated longitudinal queries $\mathbf{Q}_l^{0:T}$ from path-aware and contextual interactions, we enhance them using path-aligned reference points $\mathbf{P}_l$, providing spatial grounding for predicting offsets relative to the anchors. Specifically, the reference points $\mathbf{P}_l$ are encoded by an encoder $\text{MLP}_{ref}$ and fused with the queries to predict offsets:

$$\Delta\mathbf{Y}_l = \text{MLP}\left(\mathbf{Q}_l^{0:T} + \text{MLP}_{ref}(\mathbf{P}_l)\right), \quad \Delta\mathbf{Y}_l \in \mathbb{R}^{N_d \times M \times (T+1) \times 1}. \tag{15}$$

The final longitudinal displacements are obtained by adding offsets to the anchors:

$$\hat{\mathbf{Y}}_l = \mathbf{A}_l^{0:T} + \Delta\mathbf{Y}_l, \quad \hat{\mathbf{Y}}_l \in \mathbb{R}^{N_d \times M \times (T+1) \times 1}. \tag{16}$$

An auxiliary MLP head is applied to the average of $\mathbf{Q}_l^{0:T}$ over the $T+1$ time steps to predict a confidence score $\mathbf{S}_l \in \mathbb{R}^{N_d \times M \times 1}$ for candidate selection.

Our model outputs $N_d$ candidate drive paths $\hat{\mathbf{Y}}_d$ and, for each drive path, $M$ candidate longitudinal displacement sequences $\hat{\mathbf{Y}}_l$. A hierarchical selection strategy (Sun et al., 2024) chooses the candidate based on confidence scores $\mathbf{S}_l$ and $\mathbf{S}_d$. The selected planning is then tracked using PID controllers. Full implementation details are provided in the appendix B.

Table 1: Closed-loop results of planning in Bench2Drive. * denotes expert feature distillation. **Bold** and underlined numbers indicate the best performance within different expert groups.

| Method | Driving Score ($\uparrow$) | Success Rate (%) ($\uparrow$) | Driving Efficiency ($\uparrow$) | Comfort ($\uparrow$) |
|---|---|---|---|---|
| **Expert: PDM-Lite (Beißwenger, 2024)** | | | | |
| SimLingo (Renz et al., 2025) | 86.02 | 67.27 | 259.23 | 33.67 |
| **Expert: Think2Drive (Li et al., 2024a)** | | | | |
| UniAD-Base (Hu et al., 2023) | 45.81 | 16.36 | 129.21 | 43.58 |
| VAD (Jiang et al., 2023) | 42.35 | 15.00 | 157.94 | 46.01 |
| SparseDrive (Sun et al., 2024) | 44.54 | 16.71 | 170.21 | **48.63** |
| GenAD (Zheng et al., 2024) | 44.81 | 15.90 | - | - |
| DiFSD (Su et al., 2024) | 52.02 | 21.00 | 178.30 | - |
| DriveTransformer (Jia et al., 2025) | 63.46 | 35.01 | 100.64 | 20.78 |
| Hydra-NeXt (Li et al., 2025) | 73.86 | 50.00 | 197.76 | 20.68 |
| HiP-AD (Tang et al., 2025) | 86.77 | 69.09 | 203.12 | 19.36 |
| TCP-traj* (Wu et al., 2022) | 59.90 | 30.00 | 76.54 | 18.08 |
| ThinkTwice* (Jia et al., 2023b) | 62.44 | 31.23 | 69.33 | 16.22 |
| DriveAdapter* (Jia et al., 2023a) | 64.22 | 33.08 | 70.22 | 16.01 |
| AlignDrive(Ours) | **89.07** | **73.18** | **212.07** | 16.86 |

## 3.5 Loss Function

For planning tasks, we adopt a winner-takes-all strategy to determine which predictions are supervised. The winner is defined as the prediction whose corresponding anchor has the minimum $L_2$ distance from the ground truth. Other losses, including online mapping, agent detection, motion forecasting, and auxiliary tasks, follow (Sun et al., 2024). The total loss is the weighted sum of all components. Details are described in the appendix.

$$\mathcal{L} = \lambda_{map}\mathcal{L}_{map} + \lambda_{det}\mathcal{L}_{det} + \lambda_{motion}\mathcal{L}_{motion} + \lambda_{drivepath}\mathcal{L}_{drivepath} + \lambda_{plan}\mathcal{L}_{plan} + \lambda_{aux}\mathcal{L}_{aux}. \quad (17)$$

## 4 Experiments

### 4.1 Dataset and Metrics

**Dataset**. We utilize the Bench2Drive (Jia et al., 2024) benchmark for comprehensive evaluation of our model. This dataset consists of 1000 short video clips uniformly sampled from 44 interactive scenarios in CARLA v2 (Dosovitskiy et al., 2017). Following the official split, we use 950 clips for training and 50 for validation. Closed-loop performance is assessed on 220 standardized test routes to ensure a fair and reproducible comparison. To further assess open-loop performance in the real world, we also conduct experiments on the nuScenes dataset (Caesar et al., 2020), which consists of 1000 real-world driving scenes split into 700 for training, 150 for validation, and 150 for testing.

**Metrics.** We report the official metrics: Driving Score (DS), Success Rate (SR), Driving Efficiency (DE), and Comfort. In addition, we introduce a Collision Rate metric, defined as the proportion of scenarios involving collisions with dynamic vehicles, to specifically assess the model's capability in handling interactive environments. For open-loop evaluation, we adopt the standard metrics commonly used in prior work (Jiang et al., 2023), namely L2 distance and collision rate.

### 4.2 Implementation Details

The model employs 900 agent queries, 100 map queries, 6 drive path queries, and 5 longitudinal queries. The supervision signal for the drive path is derived from the ego vehicle's ground truth trajectory, sampled at 2-meter intervals. For longitudinal planning, the ground truth is defined as the displacements traveled along the trajectory at a 5Hz sampling rate. The longitudinal planning module employs five constant displacement anchors along the drive path, positioned at 0.25, 1.7, 4.0, 6.0, and 8.5 meters ahead of the current vehicle position. These anchors serve as reference points for predicting future longitudinal displacements. In practice, we set $\alpha = 0.1$, meaning that a virtual agent is inserted in 10% of training samples. Additional details are provided in the appendix.

### 4.3 Main Results

As shown in Table 1, our method achieves strong overall performance, with a Driving Score of 89.07 and a Success Rate of 73.18%, along with the highest Efficiency of 212.07. The Comfort

Table 2: Multi-Ability Results in Bench2Drive.* denotes expert feature distillation.

| Method | Ability (%) ↑ | | | | | |
|--------|------|---------|------------|-----------------|----------|--------------|
| | Mean | Merging | Overtaking | Emergency Brake | Give Way | Traffic Sign |
| UniAD-Base (Hu et al., 2023) | 15.55 | 14.10 | 17.78 | 21.67 | 10.00 | 14.21 |
| VAD (Jiang et al., 2023) | 18.07 | 8.11 | 24.44 | 18.64 | 20.00 | 19.15 |
| DriveTransformer-Large (Jia et al., 2025) | 38.60 | 17.57 | 35.00 | 48.36 | 40.00 | 52.10 |
| HiP-AD (Tang et al., 2025) | 65.98 | 50.00 | **84.44** | **83.33** | 40.00 | 72.10 |
| TCP-traj* (Wu et al., 2022) | 34.92 | 12.50 | 22.73 | 52.72 | 40.00 | 46.63 |
| ThinkTwice* (Jia et al., 2023b) | 37.48 | 13.72 | 22.93 | 52.99 | 50.00 | 47.78 |
| DriveAdapter* (Jia et al., 2023a) | 38.33 | 14.55 | 22.61 | 54.04 | 50.00 | 50.45 |
| AlignDrive | **70.06** | **75.00** | 75.56 | 75.00 | **50.00** | **74.74** |

Table 3: Open-loop planning evaluation results on the nuScenes validation dataset.

| Method | L2 (m) ↓ | | | | Collision (%) ↓ | | | |
|--------|------|------|------|------|------|------|------|------|
| | 1s | 2s | 3s | Avg. | 1s | 2s | 3s | Avg. |
| VAD-Base (Jiang et al., 2023) | 0.41 | 0.70 | 1.05 | 0.72 | 0.03 | 0.19 | 0.43 | 0.21 |
| GenAD (Zheng et al., 2024) | 0.28 | 0.49 | 0.78 | 0.52 | 0.08 | 0.14 | 0.34 | 0.19 |
| SparseDrive-S (Sun et al., 2024) | 0.29 | 0.58 | 0.96 | 0.61 | 0.01 | 0.05 | 0.18 | 0.08 |
| DriveTransformer-Large (Jia et al., 2025) | 0.16 | 0.30 | 0.55 | **0.33** | 0.01 | 0.06 | 0.15 | 0.07 |
| HiP-AD (Tang et al., 2025) | 0.28 | 0.53 | 0.87 | 0.56 | 0.01 | 0.05 | 0.15 | 0.07 |
| AlignDrive | 0.38 | 0.73 | 1.23 | 0.78 | 0.01 | 0.04 | 0.14 | **0.06** |

score is 16.86. This is due to challenging scenarios, such as pedestrian crossings and vehicle cut-ins, which occasionally require abrupt braking or steering. Therefore, comparisons of Comfort are most meaningful among methods with similar Success Rates, as such maneuvers are necessary to ensure safe and successful navigation.

We report the open-loop results in Table 3. Our method achieves the lowest collision rate, indicating stronger capability in handling dynamic interactions. Although the L2 distance is not the best, this is influenced by our data augmentation strategy, where inserting additional agents and adjusting the corresponding ground-truth trajectory can introduce discrepancies under an L2-based metric, while other approaches are more directly aligned with such supervision. Prior work has also noted that open-loop metrics may not fully reflect planning quality due to issues like distribution shift and causal confusion (Zhai et al., 2023; Li et al., 2024b; Dauner et al., 2023). Consistent with this, our method achieves SOTA performance in the closed-loop CARLA evaluation, which offers a more faithful measure of real-world driving behavior.

We also report the multi-ability scores in Table 2. Our model achieves the highest overall performance, with a significantly superior average score. Most notably, it reaches a Merging score of 75 far surpassing the previous best of 50. Since merging scenarios involve challenging interactions such as consecutive lane changes and cut-ins, the improvement highlights our model's enhanced capability in handling dynamic interactions and avoiding collisions, directly validating our claim.

In addition to planning ability, we evaluate inference efficiency in Table 4. Our method achieves the best Driving Score and Success Rate while maintaining lower latency than DriveTransformer and VAD. By reducing number of stakced blocks, we further develop AlignDrive-Small, which is smaller and faster than HiP-AD yet still delivers superior performance, striking a better balance between accuracy and efficiency.

Table 4: Comparison of inference efficiency and driving performance. AlignDrive-Small is a lightweight variant with fewer decoder layers. Experiments are conducted on an RTX 3090 GPU.

| Method | Parameters | Latency | Driving Score | Success Rate (%) |
|--------|-----------|---------|---------------|------------------|
| VAD-Base (Jiang et al., 2023) | - | 224.3 ms | 42.35 | 15.00 |
| DriveTransformer-Large (Jia et al., 2025) | 646 M | 221.7 ms | 63.46 | 35.01 |
| HiP-AD (Tang et al., 2025) | 97.4 M | 138.9 ms | 86.77 | 69.09 |
| AlignDrive | 117.2 M | 177.5 ms | **89.07** | **73.18** |
| AlignDrive-Small | **83.7 M** | **124.5 ms** | _87.45_ | _71.82_ |

Table 5: Ablation study on AlignDrive components. LP: uses lateral path prediction to condition longitudinal planning; DP: formulates longitudinal planning as displacement regression along the drive path; DA: applies planning-oriented data augmentation

| Variant | LP | DP | DA | Driving Score ↑ | Success Rate (%)↑ | Collision Rate (%) ↓ |
|---------|----|----|----|-----------------|-------------------|----------------------|
| A | | | | 83.21 | 63.18 | 22.7 |
| B | ✓ | | | 84.85 | 65.45 | 19.5 |
| C | ✓ | ✓ | | 85.82 | 66.81 | 16.3 |
| D | ✓ | | ✓ | 86.54 | 68.92 | 15.7 |
| E | ✓ | ✓ | ✓ | **89.07** | **73.18** | **11.4** |

Figure 4: Effect of planning-oriented data augmentation on planning performance. All augmented variants ($p = 0.1, 0.2, 0.3, 0.4$) outperform the no-augmentation baseline.

## 4.4 ABLATION STUDY

In this section, we perform ablation studies to verify the effectiveness of the key components proposed in AlignDrive, directly corresponding to our contributions.

**Independent vs Path-Conditioned Longitudinal Planning.** We compare Variant A, which predicts lateral drive path and longitudinal trajectories in parallel following prior SOTA methods (Tang et al., 2025), with Variant C, our proposed approach that predicts longitudinal displacements along the drive path. This cascaded, path-conditioned design couples lateral and longitudinal planning, resulting in more consistent and effective planning. As shown in Tab. 5, Variant C achieves a higher overall driving score and increases the Success Rate from 63.18% to 66.81%, demonstrating the effectiveness of path-conditioned longitudinal planning. In addition, Variant C reduces the Collision Rate from 22.7% to 16.3%, a 28.2% relative reduction. This improvement supports our claim that allow longitudinal planning condition on the drive path ensure the model to better focus on dynamic interactions, improving collision avoidance in complex scenarios.

**Displacement vs Waypoint Prediction.** We also evaluate Variant B, which predicts trajectory waypoints conditioned on the drive path at discrete future time steps, rather than predicting longitudinal displacements. Although both variants leverage the drive path as a lateral prior, displacements are more directly associated with dynamic interactions, whereas trajectory waypoints embed additional lateral variations that may dilute this focus. Tab. 5 shows that Variant C achieves higher Success Rate and lower Collision Rate, demonstrating that our displacement regression along the drive path is not only conceptually simpler but also empirically superior.

**Planning-Oriented Data Augmentation.** We evaluate the effectiveness of our planning-oriented data augmentation, which inserts synthetic traffic participants and adjusts longitudinal labels while keeping lateral paths unchanged. Variant C without augmentation is compared to Variant E with augmentation. As shown in Tab. 5, augmentation improves overall Driving Score from 85.82 to 89.07 and increases the Success Rate, demonstrating the effectiveness of our strategy. In addition, it reduces the Collision Rate from 16.3% to 11.4%, highlighting that augmentation helps the model better handle dynamic agents and improve safety.

Fig. 4 further illustrates the impact of augmentation across different scenarios, showing that performance slightly declines when the augmentation probability exceeds 0.1, as excessive augmentation may encourage overly conservative driving. Overall, all augmented variants substantially outperform the no-augmentation baseline, demonstrating the benefit of our strategy.

**Displacement Formulation Better Fits Augmentation.** We investigate how planning-oriented data augmentation interacts with different longitudinal representations. As shown in Tab. 5, applying augmentation to waypoint-based planning (Variant D) improves the Driving Score from 84.85 to 86.54 (+1.69), Success Rate from 65.45% to 68.92% (+3.47), and reduces Collision Rate from 19.5% to 15.7% (-3.8). In contrast, augmentation paired with displacement-based planning (Variant E) boosts

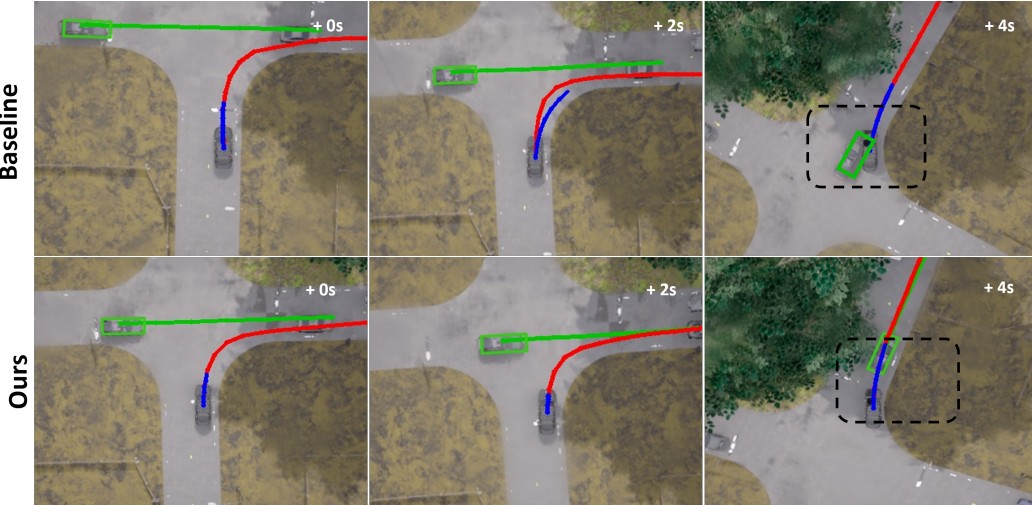

Figure 5: Red points are predicted drive paths, while blue points show longitudinal planning outputs (trajectory waypoints for the baseline, displacement squences for ours). Relevant vehicles are highlighted in green. The baseline collides with cross-traffic, while our method avoids it.

the Driving Score from 85.82 to 89.07 (+3.25), Success Rate from 66.81% to 73.18% (+6.37), and lowers Collision Rate from 16.3% to 11.4% (-4.9). These results indicate that longitudinal displacement formulation better leverages augmentation, yielding larger gains in safety-critical scenarios.

### 4.5 QUALITATIVE RESULTS

To better illustrate the effectiveness of our design, we compare our model with the baseline, which predicts drive path and trajectory independently (Variant A in Tab. 5). As shown in Fig. 5, we present a multi-vehicle interaction at an intersection where the ego vehicle must turn right while yielding to cross-traffic. In the first row, the baseline fails to react to the incoming vehicle (highlighted in purple), leading to conflict and eventual collision. In contrast, our model correctly anticipates the cross-traffic, waits until it passes, and then executes the turn safely. We provide more visulisation in the supplementary material.

## 5 CONCLUSION

We propose AlignDrive, a novel cascaded planning paradigm in which longitudinal planning is explicitly conditioned on predicted drive paths. This paradigm tightly couples lateral and longitudinal reasoning by using the path geometry as a prior for longitudinal planning. Building on this, we reformulate longitudinal planning as 1D displacement prediction along the drive path, allowing the model to focus on dynamic interactions rather than redundantly encoding static geometry. Leveraging this formulation, we introduce a planning-oriented data augmentation strategy that generates diverse, safety-critical training scenarios. Extensive evaluations show that AlignDrive achieves state-of-the-art performance, with ablation studies confirming the contribution of each component.

## REPRODUCIBILITY STATEMENT

To ensure the reproducibility of the results presented in this paper, we provide detailed descriptions of our methods and experimental setup within the main text and appendix. In addition, the supplementary material includes detailed results of closed-loop evaluations for all scenarios, and the appendix provides results from multiple simulation runs.

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

SUPPLEMENTARY MATERIAL

We include the following supplementary content:

- Additional visualization demos: a set of videos demonstrating the effectiveness of our method (see the included `index.html` file).
- Original simulation results in the CarlaV2 simulator: detailed scores for each scenario (see `AlignDrive_Meraged_Bench2drive_Results.json`).

APPENDIX

- Additional experiments, including further ablation studies.
- Detailed model designs, with full training parameters and algorithmic specifications.

## A  USE OF LARGE LANGUAGE MODELS FOR MANUSCRIPT PREPARATION.

We employed a large language model (LLM) solely to assist in language refinement and writing clarity. The LLM was **not** used for any experimental design, data analysis, or model development, and all technical content, results, and conclusions originate entirely from the authors.

## B  IMPLEMENTATION DETAILS

### B.1  TRAINING DETAILS

During training, the Longitudinal Planning module is initially frozen while the Drive Path predictor is trained for 12 epochs. The Longitudinal Planning module is then unfrozen, and the entire system is trained jointly, with the full training process spanning 36 epochs. Training is conducted on 32 NVIDIA RTX 4090 GPUs with a total batch size of 256. We use the AdamW optimizer with weight decay and set the initial learning rate to $1 \times 10^{-4}$. Planning-oriented data augmentation is introduced after 24 epochs to enrich interactive scenarios with virtual agents.

Our model predicts the next $T = 15$ drive path waypoints $\{\hat{\mathbf{Y}}_d^t\}_{t=1}^T$ at 2-meter intervals and longitudinal displacements $\{\hat{\mathbf{Y}}_l^t\}_{t=1}^T$ at 5 Hz. Supervision is applied using a weighted L1 loss:

$$\mathcal{L}_{\text{drivepath}} = \sum_{t=1}^T w_t^{\text{DP}} \|\hat{\mathbf{Y}}_d^t - \mathbf{Y}_d^t\|_1, \tag{18}$$

$$\mathcal{L}_{\text{plan}} = \sum_{t=1}^T w_t^{\text{long}} |\hat{\mathbf{Y}}_l^t - \mathbf{Y}_l^t|, \tag{19}$$

where the weights assign higher importance to more critical predictions. For the Drive Path waypoints, closer points receive larger weights: $w_t^{\text{DP}} = 1.0$ for $t = 1-5$, 0.6 for $t = 6-11$, and 0.4 for $t = 12-15$. A similar time-based weighting $w_t^{\text{long}}$ is applied to longitudinal displacements. This design encourages the model to prioritize predictions that are most critical for immediate planning and safe driving.

The weights for each component of the training objective are set as follows: $\lambda_{\text{map}} = 1$, $\lambda_{\text{det}} = 1$, $\lambda_{\text{motion}} = 1$, $\lambda_{\text{drivepath}} = 2$, $\lambda_{\text{plan}} = 2$, and $\lambda_{\text{aux}} = 1$.

### B.2  MODEL ARCHITECTURE

We implement our model using a ResNet-50 backbone  (He et al., 2016) with an input image size of $640 \times 352$. Target waypoints and high-level commands are encoded into plan queries via an MLP. During training, noise is injected into the target waypoints and commands with a certain probability to improve robustness. In the standard version of our model, we employ L = 6 layers in the Drive Path Predictor and K = 6 layers in the Longitudinal Planning module. For the AlignDrive-Small variant, we use L = 4 and K = 3.

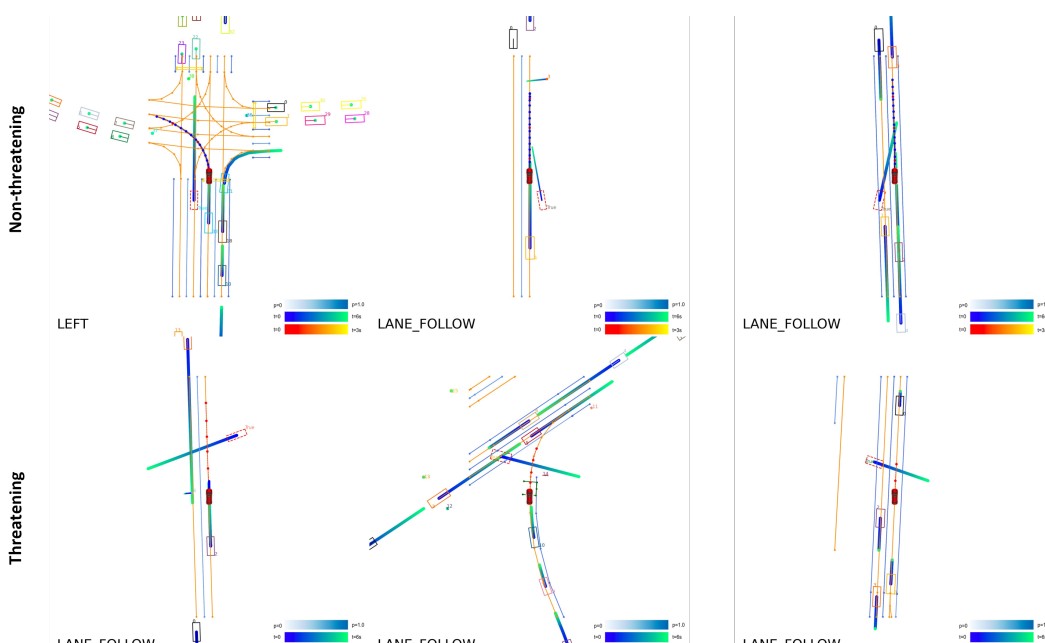

Figure 6: Visualization of planning-oriented data augmentation. The top row shows non-threatening agents, while the bottom row shows threatening agents. Inserted synthetic agents are indicated with dashed boxes. Red points denote the ego vehicle's original trajectory, and blue lines represent the adjusted longitudinal displacements after augmentation.

### B.3 PLANNING-ORITEND DATA AUGMENTATION

**Agent Insertion.** Our planning-oriented data augmentation begins with the insertion of synthetic agents (see Algorithm 1). For each training frame, we first compute the ego vehicle's displacement over the next 3 seconds. If this displacement is below a predefined threshold $\delta$, indicating that the ego vehicle is effectively stationary or moving very slowly, no augmentation is performed for that frame (line 10 of Algorithm 1).

For frames satisfying the displacement criterion, synthetic agents are inserted via a two-step process: selecting an initial position and generating a trajectory. The initial position depends on the agent type: threatening agents are sampled near the ego vehicle, while non-threatening agents are sampled from a distant range. Trajectories are determined by three parameters: the starting position $\mathbf{p}_{start}$, a waypoint $\mathbf{w}$ along the ego vehicle's future drive path, and the arrival time $t_{arrival}$ at the waypoint. Assuming constant velocity, the agent's position at each timestep is computed along the straight-line path connecting the start and waypoint. For threatening agents, the arrival time is chosen to potentially induce a collision, whereas non-threatening agents have arrival times that avoid interference. This formulation allows continuous modeling of interactions between the ego vehicle and synthetic agents.

**Displacement Ground-truth Generation.** With the synthetic agent trajectory inserted, we adjust the ego vehicle's longitudinal ground-truth displacements to ensure safety (see Algorithm 2). Specifically, we measure the distance between the ego's predicted positions and the agent at each future timestep within 3 seconds. The last point that satisfies the minimum safety distance is chosen as the new terminal point. We then compute a scaling factor as the ratio between the ego's travel distance to the adjusted terminal point and the distance to the original terminal point. This factor is used to proportionally shrink the longitudinal displacements between consecutive waypoints, preserving trajectory smoothness while guaranteeing collision-free behavior. The effectiveness of this relabeling procedure is demonstrated in our ablation results (see Table 5).

Figure 6 illustrates our planning-oriented data augmentation. The top row presents non-threatening agents, while the bottom row shows threatening agents. Inserted synthetic agents are highlighted with dashed boxes. Red points indicate the ego vehicle's original trajectory, and blue lines show the adjusted longitudinal displacements after augmentation.

---

**Algorithm 1** Planning-Oriented Agent Insertion

---

1: **Input:**
2:     $\mathbf{Y}_{ego}^{Traj}$: Ego vehicle future trajectory
3:     $\mathbf{Y}_{ego}^{DrveiPath}$: Ego vehicle future trajectory
4:     $\delta$: Displacement threshold
5:     $\alpha$: Insertion probability
6: **Output:**
7:     $\hat{\mathbf{Y}}_a^{motion}$: Synthetic agent trajectories
8: **for** each training frame **do**
9:     $D \leftarrow$ ComputeEgoDisplacement($\mathbf{Y}_{ego}^{future}$, 3s)
10:     **if** $D < \delta$ **then**
11:         **continue**
12:     **end if**
13:     **if** Random$(0, 1) \leq \alpha$ **then**
14:         $agentRole \leftarrow$ SelectAgentRole()
15:         **if** $agentRole =$ threatening **then**
16:             $\mathbf{p}_{start} \leftarrow$ SampleNearPosition($\mathbf{Y}_{ego}^{Traj}$)
17:         **else**
18:             $\mathbf{p}_{start} \leftarrow$ SampleFarPosition()
19:         **end if**
20:     $\mathbf{w} \leftarrow$ SelectWaypoint($\mathbf{Y}_{ego}^{DrveiPath}$)
21:     $t_{arrival} \leftarrow$ SampleArrivalTime()
22:     $\hat{\mathbf{Y}}_a^{motion} \leftarrow$ GenerateTrajectory($\mathbf{p}_{start}, \mathbf{w}, t_{arrival}$)
23:     **end if**
24: **end for**

---

**Algorithm 2** Displacement Ground-truth Generation

---

1: **Input:** Ego future trajectory $\mathbf{Y}_{ego}^{future}$, synthetic agent trajectory $\hat{\mathbf{Y}}_a$, minimum safe distance $d_{safe}$
2: **Output:** Adjusted ego trajectory $\mathbf{Y}_{ego}^{adjusted}$
3: Determine all future timesteps where ego is at least $d_{safe}$ away from the inserted agent
4: Let $t_{new}$ be the last safe timestep
5: Set $P_{new}$ as the ego position at $t_{new}$ (new 3s terminal point)
6: Compute scaling factor $s =$ (distance from start to $P_{new}$) / (distance from start to original terminal point)
7: **for** each consecutive pair of ego future trajectory points **do**
8:     Scale the longitudinal displacement between the points by $s$
9: **end for**
10: $\mathbf{Y}_{ego}^{adjusted} \leftarrow$ updated ego trajectory with scaled displacements

---

It is worth noting that our agent insertion relies on minimal rule-based constraints and does not explicitly use road information. As a result, the trajectories of inserted agents may violate road rules. However, this does not negatively impact the longitudinal planning module, which primarily learns to reason about potential interactions with dynamic objects rather than strict road compliance. During training, further constraining inserted agents according to road elements represents a natural extension and a promising direction for future exploration.

### B.4 AUXILIARY TASKS

We employ two primary auxiliary tasks to improve model learning. The first is ego-status prediction. Specifically, an MLP is used to predict the current ego-status of the vehicle from the plan queries, and supervision is applied using an L2 loss. The second task is inspired by the multi-granularity waypoint prediction used in HiP-AD (Tang et al., 2025). In the Drive Path Predictor, we introduce three additional types of queries that interact with the perceived environment in parallel with the drive path query. An Align-fusion strategy (Tang et al., 2025) is then applied, followed by sep-

arate heads to predict: (i) spatial waypoints at 5-meter intervals, (ii) temporal waypoints at 5Hz, and (iii) temporal waypoints at 2Hz. Each prediction is supervised independently. These auxiliary predictions are used only during training and do not participate in inference.

### B.5 SELECTION AND CONTROL

**Selection** The framework produces $N_d$ candidate drive paths and, for each drive path, $M$ longitudinal displacement sequences, representing $N_d \times M$ multimodal predictions that capture both lateral and longitudinal variations. First, the drive path with the highest confidence score $\mathbf{S}_d$ predicted by the Drive Path Predictor is selected, along with its corresponding longitudinal displacement candidates $\hat{\mathbf{Y}_1}' \in \mathbb{R}^{M \times (T+1) \times 1}$. These candidates are further scored $\mathbf{S}_l$, penalizing those that would lead to collisions with predicted motions of other agents, following SparseDrive (Sun et al., 2024). The candidate with the highest adjusted score is then chosen as the final output for downstream control. Importantly, we apply the same strategy to all variants to ensure a fair comparison in ablation studies.

**Control.** The selected candidates are executed using two independent PID controllers: one for steering and one for speed. The steering controller computes the desired heading based on the selected drive path, while the speed controller computes the desired velocity from the longitudinal displacements. Control signals for the vehicle—throttle, brake, and steering angle—are then calculated based on the difference between the desired and the current vehicle states.

Table 6: Ablation on longitudinal planning (LP), agent query decoding–re-encoding (RE), and planning-oriented data augmentation (DA). Decouple: no LP; LP + Original: LP with original agent queries; LP + Reencode: LP with decoded–re-encoded queries; Full (AlignDrive): LP + Reencode + DA.

| Method | LP | RE | DA | Driving Score ↑ | Success Rate (%) ↑ | Collision Rate (%) ↓ |
|---|---|---|---|---|---|---|
| Decouple | | | | 83.21 | 63.18 | 22.7 |
| LP + Original | ✓ | | | 87.47 | 68.18 | 15.4 |
| LP + Reencode | ✓ | ✓ | | 85.82 | 66.81 | 16.3 |
| Full (AlignDrive) | ✓ | ✓ | ✓ | **89.07** | **73.18** | **11.4** |

## C MORE EXPERIMENTS

**Effect of Re-encoding Agent Queries.** Our planning-oriented augmentation requires agent queries to be decoded into bounding boxes and then re-encoded as structured features, which enables the insertion of synthetic agents. This design differs from directly attending to the original agent queries in the longitudinal planning (LP) module, and could potentially affect performance. To futher disentangle these factors, we compare four variants: (i) **Decouple**, which excludes LP and predicts lateral and longitudinal trajectories independently; (ii) **No-Reencode**, which introduces LP but directly attends to original agent queries without decoding and re-encoding; (iii) **Reencode**, which uses LP with decoded–re-encoded agent features but without augmentation; and (iv) **Full** (AlignDrive), which combines LP, re-encoding, and planning-oriented augmentation.

As shown in Tab. 6, introducing LP (No-Reencode) already improves Driving Score and Success Rate over Decouple, demonstrating that conditioning longitudinal planning on the drive path is effective. Comparing No-Reencode and Reencode reveals a trade-off: directly using original agent queries yields stronger immediate interactions with dynamic agents, but re-encoding is necessary to support augmentation. With augmentation enabled, the Full model achieves the best overall performance, reducing collision rate most significantly, which confirms that data augmentation and displacement-based LP complement each other in improving robustness, particularly in safety-critical scenarios.

**Experimental Reproducibility.** Due to the inherent stochasticity in the CARLA closed-loop simulator, the results of a single run may slightly. To provide a more comprehensive and reliable reference, we report multiple simulation runs of our base model and compute their average performance in Tab 7. Despite these fluctuations, all runs consistently achieve state-of-the-art results, demon-

Table 7: Multiple simulation runs of AlignDrive on Bench2Drive benchmarks. Driving Score, Success Rate, Driving Efficiency, and Comfort are reported for each run along with the average.

| Run | Driving Score ↑ | Success Rate (%) ↑ | Driving Efficiency ↑ | Comfort ↑ |
|---|---|---|---|---|
| Run 1 | 89.07 | 73.18 | 212.07 | 16.86 |
| Run 2 | 87.80 | 71.36 | 207.85 | 15.25 |
| Run 3 | 88.05 | 70.00 | 210.08 | 17.10 |
| **Average** | 88.30 | 71.50 | 210.00 | 16.40 |

strating the robustness of our approach. This protocol ensures that the reported performance is representative and not an artifact of random variations in the simulation environment.

## D VISULIZATION

To further demonstrate the effectiveness of our model, we consider a scenario where a pedestrian suddenly emerges onto the road. We compare the baseline model, which predicts drive path and trajectory independently (Variant A in Table 5), with our approach. The baseline fails to react properly, resulting in a severe safety incident Fig. 7(a), whereas our method promptly responds to the pedestrian and successfully avoids the accident Fig. 7(b). More visualization video demonstrations can be found in the attached folder.

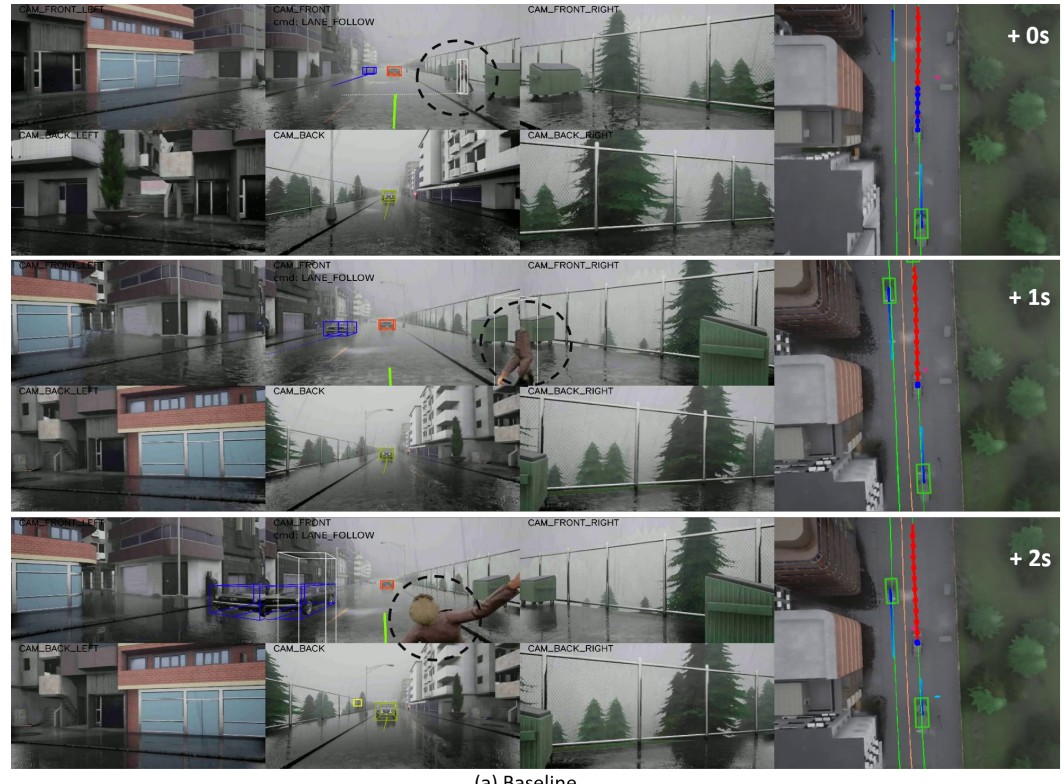

(a) Baseline

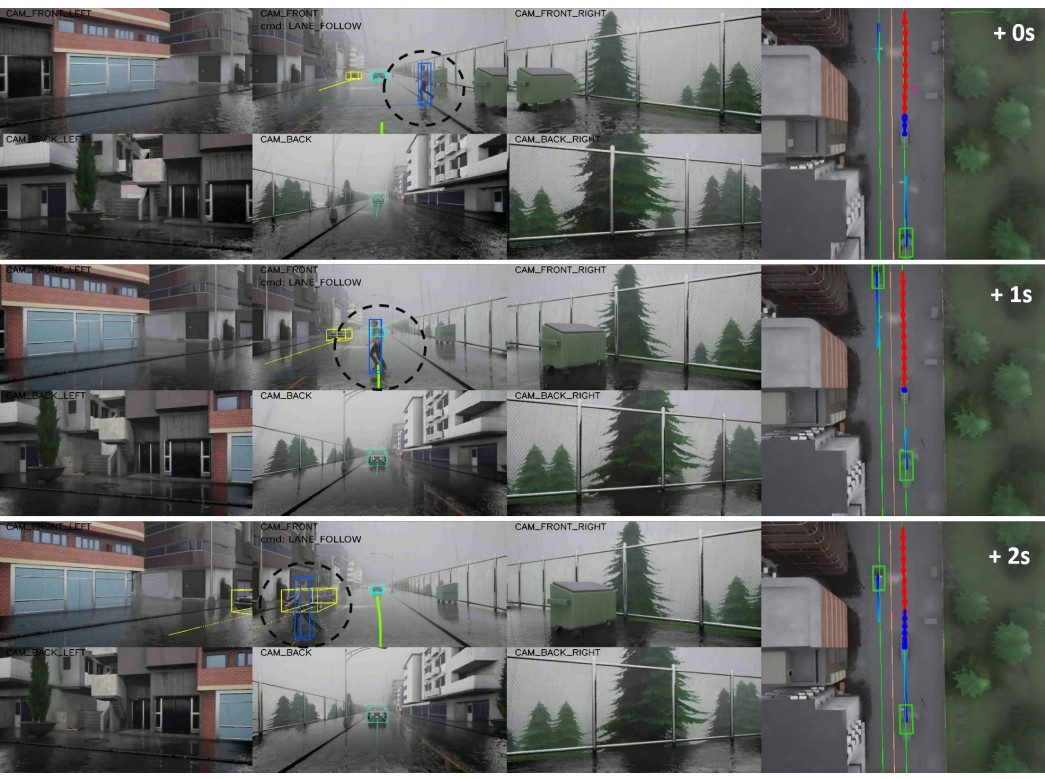

(b) Ours

Figure 7: Comparison of Baseline (a) and Ours (b) in a pedestrian cut-in scenario. The baseline model fails to avoid the pedestrian, resulting in a collision, whereas our method promptly reacts and avoids the accident. The pedestrian is highlighted with a black dashed circle.

