# OpenReview forum: "AlignDrive: Aligned Lateral-Longitudinal Planning for End-to-End Autonomous Driving"
_ICLR.cc/2026/Conference — Submitted to ICLR 2026_

### Official Review · Reviewer_zjeJ · 2025-10-16

**Soundness:** 3
**Presentation:** 2
**Contribution:** 3
**Rating:** 4
**Confidence:** 5

**Summary:**

This paper proposes a novel end-to-end autonomous driving planning model called AlignDrive. Unlike previous approaches that jointly predict lateral and longitudinal displacements, this work first predicts the driving path, and then estimates the offset along this path. This design effectively decouples path prediction from speed prediction, leading to impressive performance improvements on the Bench2Drive leaderboard. In addition, the paper also proposes a data enhancement strategy that can supplement the number of strong interaction scenarios by adding virtual agents on the driving path. This strategy also effectively improves the planning performance.

**Strengths:**

1. Unlike previous works that jointly predict lateral and longitudinal displacements, this paper introduces a trajectory planning approach that decouples path prediction from speed prediction, which is a well-motivated and effective idea.
2. On the Bench2Drive leaderboard, the proposed model demonstrates significant improvements in planning performance compared to prior methods.
3. The paper and supplementary materials provide detailed implementation information, making it easier for readers to understand and evaluate the effectiveness of the proposed approach.

**Weaknesses:**

1. Regarding data augmentation, there is an important issue that remains unclear. Why is agent encoding (Line 251) necessary? Is it because the data augmentation is not performed during data generation (i.e., the corresponding surround-view images are unchanged), and thus the model needs this encoding step to recognize the newly added agents? If so, the feature representations of these synthetic agents would inherently differ from those detected by the perception module. A better solution might be to introduce such augmentation directly during data generation, which would eliminate the need for extra encoding operations and ensure feature consistency across agents.
2. The proposed data augmentation strategy is somewhat effective, but it also has clear limitations. It only generates agents randomly along the ego vehicle’s driving path. Therefore, the generalizability of this augmentation method to broader datasets and real-world driving scenarios remains to be validated.
3. The statements in Line 015 and Line 046: “splitting planning into two independent branches can lead to coordination failures. For example, the longitudinal trajectory may indicate a high desired speed, but executing this speed along the lateral drive path can cause collisions” are not entirely accurate. Both the joint prediction of lateral and longitudinal displacements and the proposed decoupled approach can potentially lead to prediction errors and collisions; whether collisions occur depends on the prediction accuracy itself. The current phrasing could give readers the misleading impression that the proposed method inherently avoids such issues.

**Questions:**

1. I find that Figures 1 and 2 are not presented clearly enough. In Figure 1, it is unclear what the examples on the far left and far right are intended to illustrate. My understanding is that they aim to demonstrate the advantages of the proposed approach over prior methods that rely on fixed temporal or spatial interval prediction, but the visualization is not very intuitive. In Figure 2, there are several interaction modules (e.g., “Contextual Interaction”), yet it is not clearly shown which queries are involved in each interaction. Although this is described later in the text, it would be much clearer if the figure itself explicitly visualized these connections. In addition, the meaning of the numbers 0.8, 0.1, etc. on the right-hand side is not explained. Do they represent probabilities or confidence scores? Authors should describe this in the image or caption.
2. Since the proposed approach still adopts an anchor-based design, it may be inherently constrained by the dataset’s distribution. Have the authors considered an alternative strategy that directly predicts the driving path without anchors, followed by speed prediction based on that path?

---

> ### Author Response · Authors · 2025-11-21
>
> Thanks for your acknowledgement and kind advice. Regarding your concerns, we give responses below:
>
>
> > **Q1. Design Considerations for Agent Insertion in Augmentation**
> - We appreciate the suggestion of exploring image-level augmentation. Indeed, this could be a promising complementary direction for our method, and we plan to explore this direction in future work.
>
> - In our work, we do not perform augmentation at the image level; we insert agents directly into the perceived agent set at the state level. There are two main reasons for this design choice:
>   1) Simply **modifying the images cannot guarantee that the perception module will produce agents with the desired future trajectories**. In contrast, our state-level formulation enables precise motion control for inserted agents, which is essential for planning-oriented augmentation.
>   2) Directly inserting agents into images still **faces challenges in realism and efficiency**. Image-level augmentation can be computationally expensive, especially for large-scale training data generation.
> - In our agent encoding module, **both perceived and inserted agents are represented using their bounding-box states and motion attributes**, which are projected into a shared latent space via a shared MLP. Since the bounding-box states consist of interpretable properties such as position and speed, inserted agents can be seamlessly integrated with perceived agents, **ensuring a coherent representation across all agents**.
>
>
> > **Q2. Generalizability to broader real-world datasets**
>
> - In our current implementation, agents are inserted along the ego vehicle’s path. This is primarily an implementation choice rather than a limitation of the framework, which can be readily extended to more sophisticated generation strategies, such as leveraging road topology or traffic context for more realistic agent placement.
>
> - Moreover, the main purpose of our augmentation is to **provide safety-critical interactions, such as cut-ins**, which are inherently rare and difficult to capture in real-world driving logs. **Naturalistic datasets largely consist of uneventful trajectories with limited agent interactions**, making it challenging for planners to learn robust behaviors solely from raw data.
> To validate our approach, **we added experiments on the real-world dataset** nuScenes, and we observed that **our method reduces collision rates** in Table 1. Together, these results demonstrate that our approach is effective and applicable in real-world scenarios.
>
> **Table 1. Open-loop planning results on the nuScenes validation set**
> | Method | Coll 1s | Coll 2s | Coll 3s | Coll Avg |
> |--------|----------|----------|----------|-----------|
> | VAD-Base | 0.03 | 0.19 | 0.43 | 0.21 |
> | GenAD | 0.08 | 0.14 | 0.34 | 0.19 |
> | SparseDrive-S | 0.01 | 0.05 | 0.18 | 0.08 |
> | DriveTransformer-Large | 0.01 | 0.06 | 0.15 | 0.07 |
> | HiP-AD | 0.01 | 0.05 | 0.15 | 0.07 |
> | **Our Method** | **0.01** | **0.04** | **0.14** | **0.06** |

---

> ### Author Response · Authors · 2025-11-21
>
> > **Q3. Clarification of coordination statements**
>
> - Thanks for your constructive review. We agree that collision risk primarily depends on prediction accuracy. Consistent with this, AlignDrive predicts longitudinal planning by leveraging the predicted lateral drive path as a prior, which helps the model focus more on dynamic interactions rather than static elements. **This design helps improve longitudinal planning accuracy**, as evidenced by the lower collision rate observed in our ablation experiments (Table 5).
>
> - A further consideration is that prior decoupled approaches often predict longitudinal and lateral planning independently, implicitly assuming that the longitudinal trajectory will naturally align with the lateral path. Deviations from this assumption can lead to inconsistent predictions. For instance, as illustrated in the top-right of Fig. 1(b), the longitudinal trajectory predicted independently may be collision-free on its own, but applying its speed along a separately predicted lateral path could cause a collision. By explicitly conditioning longitudinal planning on the predicted lateral drive path, AlignDrive improves consistency between lateral and longitudinal decisions, thereby reducing the risk of such inconsistencies.
>
> - We admitted that the risk of collision is a matter of prediction accuracy in any model, not an inherent flaw in the prior decoupled architecture itself. **We would like to shift from the guaranteed collision to the difficulty of learning consistency** and suboptimal resource allocation in the independent-branch approach. To clarify the original text in Lines 015 and 046, we will revise it to: “Splitting planning into two independent branches makes it difficult to enforce kinematic consistency between the resulting outputs. For example, as shown in the top-right of Fig. 1(b), a lateral path requiring a sharp turn and a longitudinal trajectory demanding high speed are not constrained to be mutually consistent during training, potentially leading to inconsistent predictions that challenge downstream execution".
>
>
> > **Q4. Visualization quality**
>
> - We apologize for the confusion caused by the original figures and have updated Figures 1 and 2 accordingly. The revised captions now provide clearer and more detailed explanations. In particular, in Figure 2, the black numbers indicate the scores of the predicted drive paths, while the red numbers denote the corresponding longitudinal-planning scores associated with each drive path. These scores are used to select the final output among the multimodal predictions. We hope these clarifications improve the readability of the visualizations and sincerely appreciate the reviewer’s helpful feedback.
>
>
> > **Q5. Anchor-free model performance**
>
> - In the original setup, the model uses dataset derived anchors to initialize learnable embeddings, which serve as references for predicting the drive path. Following the reviewer’s suggestion, we conducted an experiment in which these embeddings are randomly initialized during training. As shown in the table 2, this variant exhibits a performance drop.  In addition, we include a variant (ParallelPlanner) that predicts the lateral drive path and longitudinal trajectories in parallel, following prior methods, and it similarly exhibits a noticeable degradation. We believe this is because the dataset derived anchors provide a strong initialization that stabilizes optimization and improves training efficiency. This aligns with the broader design practice in many existing methods [1, 2], where anchor-based formulations are commonly employed to provide stable structural priors and facilitate optimization.
> - Although the current results favor anchor-based designs, we expect anchor-free designs to become more competitive as the dataset scales, which we plan to investigate in future work.
>
>
> **Table2: Closed-loop results on Bench2Drive.**
>
> | Method                     | Driving Score (↑) | Success Rate (%) (↑) |
> |----------------------------|-----------------|--------------------|
> | SparseDrive                | 44.54           | 16.71              |
> | ParallelPlanner (anchor-free) | 77.95           | 54.09
> | ParallelPlanner               | 83.21           | 63.18
> | AlignDrive (anchor-free)   | 84.24           | 65.45              |
> | AlignDrive                  | **89.07**       | **73.18**          |
>
>
>
>
> [1] Yingqi Tang, Zhuoran Xu, Zhaotie Meng, and Erkang Cheng. Hip-ad: Hierarchical and multi-granularity planning with deformable attention for autonomous driving in a single decoder. Proceedings of the IEEE/CVF International Conference on Computer Vision (ICCV), 2025, pp. 25605-25615
>
> [2] Liao, Bencheng, et al. "Diffusiondrive: Truncated diffusion model for end-to-end autonomous driving." Proceedings of the Computer Vision and Pattern Recognition Conference. 2025.

---

> > ### Comment · Reviewer_zjeJ · 2025-11-27
> >
> > Thank you for the detailed responses, which resolved most of my concerns. However, I still have one remaining doubt regarding Weakness 1. The authors explained the motivation for using agent encoding, but in actual inference, when an emergency occurs (e.g., cut in), there will be noticeable distributional differences in the image (which are not modified by the data augmentation). As a result, the features obtained through augmentation will deviate from those in real scenarios. Moreover, the collision-rate metric on nuScenes is already close to saturation, so it provides limited reference value. In addition, the L2 metric on nuScenes shows a clear gap compared with prior models.

---

> > > ### Author Response · Authors · 2025-11-27
> > >
> > > We would greatly appreciate your timely feedback on the following clarifications regarding:
> > >
> > > > **Q1. Clarification on data augmentation**
> > >
> > > We appreciate the reviewer’s concern regarding distribution shifts in emergency scenarios, which our design explicitly aims to mitigate. To further clarify, as indicated in Section 3.4, **our longitudinal planning module does not use raw image features**; instead, it relies entirely on encoded agent states from the agent encoding module together with the driving path and map queries. By depending only on structured agent-level information, the planner remains robust to changes in visual appearance during rare events such as cut-ins.
> > >
> > > > **Q2. Effect of data augmentation**
> > >
> > > - To further evaluate the effect of our data augmentation strategy, we conducted an ablation study by disabling the data augmentation. As shown in Table 1,**disabling data augmentation leads to a significant drop in collision rate (a 33% drop in safety)**. This provides clear evidence that, even though the collision-rate metric on nuScenes is close to saturation, our strategy still brings substantial improvements in real-world safety, demonstrating its practical value.
> > >
> > >
> > > > **Q3. Limitations of the open-loop L2 metric**
> > >
> > > - We acknowledge that the L2 metric on nuScenes shows a gap compared with prior models. However, **open-loop L2 metrics have well-recognized limitations and do not fully reflect real-world driving performance [1–4]**. In particular, Wayve [5] demonstrated in real-world experiments that **better open-loop metrics (e.g., L1) do not ecessarily correspond to better real-world driving performance**. This reflects the fact that real-world driving is inherently reactive and closed-loop, so trajectory metrics cannot fully capture actual driving quality.
> > >
> > > - The observed increase in L2 for our method arises from our planning-oriented data augmentation strategy. This augmentation intentionally introduces additional safety-critical agents and adjusts the supervised trajectories to **teach model collision-aware driving behavior, such as modulating acceleration and deceleration to safely avoid potential collisions**. This necessary **shift moves the trajectories away from the original distribution**, whose expert trajectories involve only limited agent interactions, naturally leading to a higher L2 when evaluated against the original ground truth.
> > >
> > > Together with reactive closed-loop evaluation in Bench2Drive, experiments on nuScenes offer complementary validation on real-world driving data, supporting the robustness and safety of our approach
> > >
> > >
> > > **Table 1 : Open-loop planning evaluation results on the nuScenes validation dataset. DA indicates planning-oriented data augmentation.**
> > > | Method | || **L2 (m) ↓** ||  | | | | | | | | | **Collision (%) ↓** |  | | |
> > > |:-------|:-------------:|:-------------:|:-------------:|:-------------:|:-------------:|:-------------:|:-------------:|:-------------:|:-------------:|:-------------:|:-------------:|:-------------:|:-------------:|:-------------:|:-------------:|:-------------:|:-------------:|
> > > |        | **1s** || **2s** || **3s** | | **Avg.** | | | | | | **1s** | **2s** | **3s** | | **Avg.** |
> > > | SparseDrive-S [Sun et al. 2024] | 0.29 || 0.58 || 0.96 | | 0.61 | | | | | | 0.01 | 0.05 | 0.18 | | 0.08 |
> > > | AlignDrive-w/o da | 0.32 || 0.58 || 0.94 | |**0.61** | | | | | | 0.01 | 0.06 | 0.16 | | 0.08 |
> > > | AlignDrive | 0.38 || 0.73 || 1.23 | | 0.78 | | | | | | 0.01 | 0.04 | 0.14 | | **0.06** |
> > >
> > >
> > >
> > > [1] Jiang-Tian Zhai, Ze Feng, Jinhao Du, Yongqiang Mao, Jiang-Jiang Liu, Zichang Tan, Yifu Zhang, Xiaoqing Ye, and Jingdong Wang. Rethinking the open-loop evaluation of end-to-end autonomous
> > > driving in nuscenes. arXiv preprint arXiv:2305.10430, 2023.
> > >
> > > [2] Zhiqi Li, Zhiding Yu, Shiyi Lan, Jiahan Li, Jan Kautz, Tong Lu, and Jose M Alvarez. Is ego status all you need for open-loop end-to-end autonomous driving? In Proceedings of the IEEE/CVF
> > > Conference on Computer Vision and Pattern Recognition, pp. 14864–14873, 2024b.
> > >
> > > [3] Daniel Dauner, Marcel Hallgarten, Andreas Geiger, and Kashyap Chitta. Parting with miscon-
> > > ceptions about learning-based vehicle motion planning. In Conference on Robot Learning, pp.
> > > 1268–1281. PMLR, 2023.
> > >
> > > [4] Jia, Xiaosong, et al. "Bench2drive: Towards multi-ability benchmarking of closed-loop end-to-end autonomous driving." Advances in Neural Information Processing Systems 37 (2024): 819-844.
> > >
> > > [5] Hawke, Jeffrey, et al. "Urban driving with conditional imitation learning."2020 IEEE International Conference on Robotics and Automation (ICRA). IEEE, 2020.

---

### Official Review · Reviewer_dQSN · 2025-10-29

**Soundness:** 2
**Presentation:** 2
**Contribution:** 2
**Rating:** 2
**Confidence:** 5

**Summary:**

This paper introduces AlignDrive, a cascaded planning framework for end-to-end autonomous driving. The core idea is to tightly couple lateral (path) and longitudinal (trajectory) planning by explicitly conditioning longitudinal planning on the predicted lateral path, addressing coordination failures in prior methods. The authors also reformulate longitudinal planning as a 1D displacement prediction task along the driving path, simplifying dynamic interaction modeling. Additionally, a planning-oriented data augmentation strategy is proposed to generate diverse safety-critical scenarios, improving the robustness of the model. Experiments on the closed-loop benchmark Bench2Drive demonstrate state-of-the-art performance for AlignDrive.

**Strengths:**

- Clear problem definition with practical significance.

The paper addresses the critical issue of coordination between lateral and longitudinal planning in end-to-end autonomous driving, which has direct implications for safety and performance.


- Well-structured method design.

The introduction of path-conditioned planning and 1D displacement prediction simplifies the longitudinal planning task while improving dynamic interaction modeling.

- Effective data augmentation.

The proposed planning-oriented data augmentation generates diverse safety-critical scenarios, significantly improving the model’s robustness in complex interactions.

- Strong experimental results.

AlignDrive achieves superior performance on the Bench2Drive benchmark, particularly in success rate and collision avoidance metrics.

**Weaknesses:**

- Incremental contribution.

While the proposed method introduces path-conditioned planning and displacement prediction, the overall approach is relatively incremental. Lateral and longitudinal coupling is already a common practice in traditional autonomous driving planning, limiting the novelty of this work.

- Incomplete evaluation.

Although Bench2Drive is a challenging closed-loop benchmark, there remains a significant gap between simulation results and real-world driving scenarios. The authors should evaluate their method on semi-open-loop benchmarks, such as NavSim, to better assess generalization. Closed-loop benchmarks cannot fully eliminate the possibility of designing test-specific tricks. Open-loop evaluations would help validate the robustness of the method further.

- Writing quality needs improvement.

The paper contains many sections that exhibit patterns typical of large language models. The writing lacks conciseness and academic rigor, which detracts from the overall quality. The authors should refine the language for better readability and professionalism.

**Questions:**

Please further discuss the contribution and novelty of this work.

---

> ### Author Response · Authors · 2025-11-21
>
> Thanks for your acknowledgement and kind advice. Regarding your concerns, we give responses below:
>
> > **Q1. Clarification on Novelty and Contribution**
>
> 1. Addressing lateral–longitudinal coupling
>
> - Regarding lateral–longitudinal coupling, prior work generally follows two paradigms: (1) **fully coupled trajectory regression** (e.g., SparseDrive, UniAD), which predicts only trajectories for both lateral and longitudinal controlling; (2) **decoupled parallel methods** (e.g., TF++ and HiP-AD), which predict drive paths and trajectories parallely. While these methods benefit from path priors, their longitudinal reasoning is not explicitly conditioned on the drive path, potentially leading to inconsistencies. Unlike methods that predict only trajectories or that predict the drive path and longitudinal trajectory in parallel without explicit dependency, we propose an explicitly cascaded coupling, where the drive path is predicted first, followed by longitudinal planning conditioned on this path. This sequential design ensures coherent and coordinated lateral–longitudinal predictions, improving collision avoidance and enabling more reliable handling of dynamic interactions.
>
> - **Instantiating a cascaded coupling framework is not trivial**. To address this, we first explored several non-trivial strategies. In one approach, longitudinal queries performed cross-attention over both drive path queries and environmental features for direct trajectory prediction; however, the resulting trajectories did not fully exploit the drive path prior. Another approach predicted longitudinal control signals (e.g., brake, acceleration) directly from the drive path, but its effectiveness was limited by a single output modality. Both approaches proved suboptimal, motivating our anchor-based formulation.
> Finally, we **formulate longitudinal planning as an anchor-based offset regression task**, where a set of anchors representing candidate longitudinal displacements along the predicted drive path is initialized as references. During iterative refinement, the model predicts offsets relative to these anchors. This design provides two main benefits: it allows the model to fully perceive the path shape and environment, and the diversity of anchor initialization enables mutil-modal longitudinal planning. By grounding predictions on these anchors, longitudinal planning is explicitly conditioned on the selected drive path, resulting in more coordinated and accurate planning.
>
> - Ablation studies (Table 5) under identical settings show that **our cascaded coupling consistently outperforms alternative schemes**, demonstrating that explicit conditioning yields more reliable coordination.
>
> 2. Additional contributions enabled by this design
>
> - Our formulation **expresses longitudinal planning as 1D displacements along the drive path**, strengthening dynamic-interaction reasoning and contributing to improved performance.
>
> - Based on the displacement-based cascaded planning framework, **we proposed a planning-oriented data augmentation, which explicitly targets safety-critical interactions and further enhances robustness.**.
>
> - Together, these components form a **coherent and empirically validated framework, achieving state-of-the-art performance** and going beyond incremental modifications to prior coupling paradigms.

---

> ### Author Response · Authors · 2025-11-21
>
> > **Q2. Evaluation Completeness**
>
> - To address potential concerns about tricks in closed-loop evaluation, we clarify that **our ablation studies (Table 5) are specifically designed to isolate the contribution of each module under strictly controlled conditions**, with all variants using the same training protocol, data, and evaluation setup, **ensuring fairness and eliminating benchmark-specific adaptations**.
>
> - Furthermore, we report **open-loop performance on the real-world dataset nuScenes**. Our method achieves the **lowest collision rate**, demonstrating a stronger capability to handle dynamic interactions. Although the L2 distance is not the best, this is largely influenced by our data augmentation strategy, which inserts additional agents during training and adjusts the corresponding ego trajectory supervision. This causes the learned model to be optimized toward augmented trajectories that may slightly differ from the original ground-truth trajectories, resulting in discrepancies under an L2-based metric. In contrast, other approaches remain more directly aligned with the original ground-truth supervision.
>
> - At the same time, as evidenced by prior studies [1–4], **open-loop metrics, while informative, have recognized limitations and may not fully capture real-world driving performance**. This limitation has also been underscored in industry discussions, including Tesla’s ICCV 2025 talk [5]. While open-loop evaluation offers a standardized comparison across methods, **real-world driving is inherently closed-loop and reactive**, making closed-loop evaluation the most indicative of practical planning robustness. For this reason, we regard **closed-loop assessment as the primary evidence of a framework's real-world capability**, and our method is validated accordingly in a reactive closed-loop simulator.
>
> > **Q3. Writing quality**
>
> We appreciate the feedback and will further refine the writing to improve readability and academic rigor.
>
>
> **Table 1 : Open-loop planning evaluation results on the nuScenes validation dataset.**
> | Method | || **L2 (m) ↓** ||  | | | | | | | | | **Collision (%) ↓** |  | | |
> |:-------|:-------------:|:-------------:|:-------------:|:-------------:|:-------------:|:-------------:|:-------------:|:-------------:|:-------------:|:-------------:|:-------------:|:-------------:|:-------------:|:-------------:|:-------------:|:-------------:|:-------------:|
> |        | **1s** || **2s** || **3s** | | **Avg.** | | | | | | **1s** | **2s** | **3s** | | **Avg.** |
> | VAD-Base [Jiang et al. 2023] | 0.41 || 0.70 || 1.05 | | 0.72 | | | | | | 0.03 | 0.19 | 0.43 | | 0.21 |
> | GenAD [Zheng et al. 2024] | 0.28 || 0.49 || 0.78 | | 0.52 | | | | | | 0.08 | 0.14 | 0.34 | | 0.19 |
> | SparseDrive-S [Sun et al. 2024] | 0.29 || 0.58 || 0.96 | | 0.61 | | | | | | 0.01 | 0.05 | 0.18 | | 0.08 |
> | DriveTransformer-Large [Jia et al. 2025] | 0.16 || 0.30 || 0.55 | | **0.33** | | | | | | 0.01 | 0.06 | 0.15 | | 0.07 |
> | HiP-AD [Tang et al. 2025] | 0.28 || 0.53 || 0.87 | | 0.56 | | | | | | 0.01 | 0.05 | 0.15 | | 0.07 |
> | AlignDrive | 0.38 || 0.73 || 1.23 | | 0.78 | | | | | | 0.01 | 0.04 | 0.14 | | **0.06** |
>
>
> [1] Jiang-Tian Zhai, Ze Feng, Jinhao Du, Yongqiang Mao, Jiang-Jiang Liu, Zichang Tan, Yifu Zhang, Xiaoqing Ye, and Jingdong Wang. Rethinking the open-loop evaluation of end-to-end autonomous
> driving in nuscenes. arXiv preprint arXiv:2305.10430, 2023.
>
> [2] Zhiqi Li, Zhiding Yu, Shiyi Lan, Jiahan Li, Jan Kautz, Tong Lu, and Jose M Alvarez. Is ego status
> all you need for open-loop end-to-end autonomous driving? In Proceedings of the IEEE/CVF
> Conference on Computer Vision and Pattern Recognition, pp. 14864–14873, 2024b.
>
> [3] Daniel Dauner, Marcel Hallgarten, Andreas Geiger, and Kashyap Chitta. Parting with miscon-
> ceptions about learning-based vehicle motion planning. In Conference on Robot Learning, pp.
> 1268–1281. PMLR, 2023.
>
> [4] Jia, Xiaosong, et al. "Bench2drive: Towards multi-ability benchmarking of closed-loop end-to-end autonomous driving." Advances in Neural Information Processing Systems 37 (2024): 819-844.
>
> [5] Ashok Elluswamy, Vice President, AI, Tesla Autonomy Team. “Building an Autonomous Future.” ICCV 2025 presentation, Tesla, Oct 2025. YouTube video: https://www.youtube.com/watch?v=IRu-cPkpiFk.

---

> > ### Comment · Reviewer_dQSN · 2025-11-26
> > **still have concerns**
> >
> > Thank you for the authors' feedback.
> >
> > To further clarify the technical details of your work,I have the following questions for your consideratio.
> >
> > - Could you elaborate on the reasons why the L2 metric performs suboptimally in your experimental settings? Specifically, is it related to data characteristics, model design, or other factors?
> >
> > - Again, have you considered evaluating your method on NavSim?

---

> > > ### Author Response · Authors · 2025-11-26
> > >
> > > > **Q1. Perfomrance of L2 metric**
> > >
> > > - The suboptimal performance of the L2 distance is **not related to our model design**. As shown in Table 1, when we disable data augmentation, our method achieves an L2 score comparable to SparseDrive.  The observed degradation in L2 arises from our planning-oriented data augmentation strategy.  Notably, this augmentation intentionally introduces additional safety-critical agents and adjusts the supervision trajectory to teach the model **collision-aware** behavior.  This necessary shift moves the supervised trajectories away from the original distribution, whose expert trajectories involve only limited agent interactions, leading to a higher L2 distance during evaluation against the original ground truth.
> > >
> > > - At the same time, as evidenced by prior studies [1–4], open-loop L2  metrics have well-recognized limitations and do not fully reflect real-world driving performance. Thus, although our data augmentation slightly affects the L2 metric, it contributes positively to closed-loop safety and robustness, **which are more aligned with real-world requirements**.
> > >
> > > > **Q2. Evaluation on Navsim**
> > > - We are actively conducting experiments on NavSim; however, due to time constraints, we have not yet completed them. We plan to provide these results in the final version.
> > >
> > > - It is important to emphasize that **our work targets the closed-loop and reactive driving setting**, which better reflects real-world deployment conditions. Bench2Drive is specifically designed for this setting.
> > >    1) Consistent with this protocol, all baseline methods we compare against—such as HiP-AD (ICCV 2025)[5], DriveTransformer (ICLR 2025)[6], SimLingo (CVPR 2025)[7], MomAD (CVPR 2025)[8], and SparseDrive (ICRA 2025)[9] **report results exclusively on Bench2Drive and do not evaluate on NavSim**.
> > >    2) In contrast, methods designed around a different experimental setting, primarily represented by NavSim, such as DiffusionDrive (CVPR 2025)[10], GoalFlow (CVPR 2025)[11], DistillDrive (ICCV 2025)[12], and PARA-Drive (CVPR 2024)[13], **conduct experiments only on NavSim** and do not include Bench2Drive results.
> > >
> > >    These publication patterns clearly indicate that Bench2Drive and NavSim correspond to distinct experimental regimes, and **it is therefore not standard practice to evaluate methods on both simultaneously**.
> > >
> > > - Nevertheless, following your suggestion, we will continue to carry out the NavSim experiments and report the results in the final version.
> > >
> > >
> > >
> > > **Table 1 : Open-loop planning evaluation results on the nuScenes validation dataset. DA indicates planning-oriented data augmentation.**
> > > | Method | || **L2 (m) ↓** ||  | | | | | | | | | **Collision (%) ↓** |  | | |
> > > |:-------|:-------------:|:-------------:|:-------------:|:-------------:|:-------------:|:-------------:|:-------------:|:-------------:|:-------------:|:-------------:|:-------------:|:-------------:|:-------------:|:-------------:|:-------------:|:-------------:|:-------------:|
> > > |        | **1s** || **2s** || **3s** | | **Avg.** | | | | | | **1s** | **2s** | **3s** | | **Avg.** |
> > > | SparseDrive-S [Sun et al. 2024] | 0.29 || 0.58 || 0.96 | | 0.61 | | | | | | 0.01 | 0.05 | 0.18 | | 0.08 |
> > > | AlignDrive-w/o da | 0.32 || 0.58 || 0.94 | |**0.61** | | | | | | 0.01 | 0.06 | 0.16 | | 0.08 |
> > > | AlignDrive | 0.38 || 0.73 || 1.23 | | 0.78 | | | | | | 0.01 | 0.04 | 0.14 | | **0.06** |

---

> > > ### Author Response · Authors · 2025-11-26
> > >
> > > [1] Jiang-Tian Zhai, Ze Feng, Jinhao Du, Yongqiang Mao, Jiang-Jiang Liu, Zichang Tan, Yifu Zhang, Xiaoqing Ye, and Jingdong Wang. Rethinking the open-loop evaluation of end-to-end autonomous
> > > driving in nuscenes. arXiv preprint arXiv:2305.10430, 2023.
> > >
> > > [2] Zhiqi Li, Zhiding Yu, Shiyi Lan, Jiahan Li, Jan Kautz, Tong Lu, and Jose M Alvarez. Is ego status all you need for open-loop end-to-end autonomous driving? In Proceedings of the IEEE/CVF
> > > Conference on Computer Vision and Pattern Recognition, pp. 14864–14873, 2024b.
> > >
> > > [3] Daniel Dauner, Marcel Hallgarten, Andreas Geiger, and Kashyap Chitta. Parting with miscon-
> > > ceptions about learning-based vehicle motion planning. In Conference on Robot Learning, pp.
> > > 1268–1281. PMLR, 2023.
> > >
> > > [4] Jia, Xiaosong, et al. "Bench2drive: Towards multi-ability benchmarking of closed-loop end-to-end autonomous driving." Advances in Neural Information Processing Systems 37 (2024): 819-844.
> > >
> > > [5] Yingqi Tang, Zhuoran Xu, Zhaotie Meng, and Erkang Cheng. Hip-ad: Hierarchical and multi-granularity planning with deformable attention for autonomous driving in a single decoder. Proceedings of the IEEE/CVF International Conference on Computer Vision (ICCV), 2025, pp. 25605-25615
> > >
> > > [6] Jia, Xiaosong, et al. "DriveTransformer: Unified Transformer for Scalable End-to-End Autonomous Driving." International Conference on Learning Representations (ICLR), 2025.
> > >
> > > [7] Renz, Katrin, et al. "Simlingo: Vision-only closed-loop autonomous driving with language-action alignment." Proceedings of the Computer Vision and Pattern Recognition Conference. 2025.
> > >
> > > [8] Song, Ziying, et al. "Don't Shake the Wheel: Momentum-Aware Planning in End-to-End Autonomous Driving." Proceedings of the Computer Vision and Pattern Recognition Conference. 2025.
> > >
> > > [9] Sun, Wenchao, et al. "Sparsedrive: End-to-end autonomous driving via sparse scene representation." 2025 IEEE International Conference on Robotics and Automation (ICRA). IEEE, 2025.
> > >
> > > [10] Liao, Bencheng, et al. "Diffusiondrive: Truncated diffusion model for end-to-end autonomous driving." Proceedings of the Computer Vision and Pattern Recognition Conference. 2025.
> > >
> > > [11] Xing, Zebin, et al. "Goalflow: Goal-driven flow matching for multimodal trajectories generation in end-to-end autonomous driving." Proceedings of the Computer Vision and Pattern Recognition Conference. 2025.
> > >
> > > [12] Yu, Rui, et al. "DistillDrive: End-to-End Multi-Mode Autonomous Driving Distillation by Isomorphic Hetero-Source Planning Model." Proceedings of the IEEE/CVF International Conference on Computer Vision. 2025.
> > >
> > > [13] Weng, Xinshuo, et al. "Para-drive: Parallelized architecture for real-time autonomous driving." Proceedings of the IEEE/CVF Conference on Computer Vision and Pattern Recognition. 2024.

---

### Official Review · Reviewer_GeYV · 2025-10-29

**Soundness:** 3
**Presentation:** 2
**Contribution:** 3
**Rating:** 6
**Confidence:** 3

**Summary:**

The paper introduces a new end-to-end autonomous driving framework AlignDrive, which aims at improving coordination between longitudinal and lateral planning. The approach leverages a cascaded path-conditioned formulation that tightly couples longitudinal and lateral planning through the use of anchor-based displacement regression along the drive path. Additionally, the framework incorporates a planning-oriented data augmentation strategy that simulates safety-critical events by inserting synthetic agents and adjusting longitudinal displacements. Evaluated on the challenging Bench2Drive benchmark, AlignDrive achieves state-of-the-art closed-loop driving performance, setting a new driving score of 89.07 and a success rate of 73.18%.

**Strengths:**

1. AlignDrive effectively addresses the limitations of previous methods like TF++ and HiP-AD by coordinating longitudinal and lateral planning through a novel cascaded design. This tightly couples the two tasks, ensuring better path-following and collision avoidance in complex driving scenarios. This is also the major novelty of the proposed framework.

2. The framework demonstrates impressive closed-loop driving performance, achieving a significant improvement over prior methods. It sets a new benchmark with the highest driving score and success rate on the Bench2Drive dataset.

3. The authors introduce a planning-oriented data augmentation approach that simulates rare safety-critical events like vehicle cut-ins. This strategy helps the model learn to navigate challenging interactions, ultimately improving its ability to avoid collisions in dynamic environments.

4. The paper provides detailed ablation studies that clearly demonstrate the necessity of each component in the AlignDrive framework, including the path-conditioned longitudinal planning and the data augmentation strategy. This strengthens the validity of the proposed approach.

**Weaknesses:**

1. The visual quality of Figure 1 could be improved, as some icons are not very clear. This affects the overall presentation and could confuse readers trying to follow the diagram. Additionally, Figure 5 is hard to interpret as the figure appears overly clustered.

2. While the paper compares AlignDrive with methods like HiP-AD, it does not include a comparison with other notable previous works such as TF++, SimLingo, or Hydra-Next in Table 1. Even though some of these methods use different training data, a comparative analysis would provide a more complete context for the performance claims.

3. The paper focuses primarily on the Bench2Drive dataset. It would be beneficial to test AlignDrive on other datasets, such as nuScenes or NAVSIM, to validate its ability to generalize across different environments and real-world data.

**Questions:**

Please refer to the weaknesses section.

---

> ### Author Response · Authors · 2025-11-21
>
> Thanks for your acknowledgement and kind advice. Regarding your concerns, we give responses below:
>
> > **Q1. Visual quality**
>
> Thank you for pointing this out. We have improved the visual quality of Figure 1 and clarified the icons. Figure 5 has also been refined to reduce cluster and enhance interpretability, and these updates have been incorporated into the revised manuscript.
>
> > **Q2. Compare with more methods**
>
> Thank you for the valuable suggestion. We have updated Table 1 to include comparisons with SimLingo and Hydra-NeXt. To illustrate this more clearly, we provide an excerpt of the updated table here. Notably, even with Hydra-NeXt and SimLingo included, our model still achieves state-of-the-art performance on Driving Score and Success Rate.
>
> While TF++ has not been evaluated on Bench2Drive and is therefore not included in the quantitative comparison, it is discussed in the Related Work section. We will also expand the discussion in the main text to provide a clearer context for all three methods.
>
> **Closed-loop results of planning in Bench2Drive.**
>
> | Method                         | Driving Score ($\uparrow$) | Success Rate (%) ($\uparrow$) | Driving Efficiency ($\uparrow$) | Comfort ($\uparrow$) |
> |--------------------------------|---------------------------|-------------------------------|---------------------------------|---------------------|
> | **Expert: PDM-Lite**           |                           |                               |                                 |                     |
> | SimLingo                        | _85.07_                  | _67.27_                       | _259.23_                        | _33.67_             |
> | **Expert: Think2Drive**        |                           |                               |                                 |                     |
> | SparseDrive                     | 44.54                     | 16.71                         | 170.21                          | **48.63**           |
> | Hydra-NeXt                      | 73.86                     | 50.00                         | 197.76                          | 20.68               |
> | HiP-AD                          | 86.77                     | 69.09                         | 203.12                          | 19.36               |
> | **AlignDrive**          | **89.07**                 | **73.18**                     | **212.07**                       | 16.86               |

---

> ### Author Response · Authors · 2025-11-21
>
> > **Q3. Evaluating on real real-world dataset**
>
> We have extended our evaluation to the **real-world dataset** nuScenes to test AlignDrive’s generalization **across different environments**. As shown in Table 2, our method achieves the **lowest collision rate**, demonstrating stronger handling of dynamic interaction in real-world traffic scenarios. The slightly higher L2 distance arises from our data augmentation strategy, which introduces additional agents and adjusts corresponding ground-truth trajectories during training; this can create small deviations from the original expert trajectories, which are penalized by L2 metrics.
>
> It is worth noting that such open-loop metrics may not always fully reflect planning quality [1,2,3]. In contrast, **our method consistently achieves state-of-the-art performance in the closed-loop Bench2drive benchmark**, which better **represents real-world driving conditions and dynamic interactions**.
>
> **Table 2: Open-loop planning evaluation results on the nuScenes validation dataset.**
>
> | Method | || **L2 (m) ↓** ||  | | | | | | | | | **Collision (%) ↓** |  | | |
> |:-------|:-------------:|:-------------:|:-------------:|:-------------:|:-------------:|:-------------:|:-------------:|:-------------:|:-------------:|:-------------:|:-------------:|:-------------:|:-------------:|:-------------:|:-------------:|:-------------:|:-------------:|
> |        | **1s** || **2s** || **3s** | | **Avg.** | | | | | | **1s** | **2s** | **3s** | | **Avg.** |
> | VAD-Base [Jiang et al. 2023] | 0.41 || 0.70 || 1.05 | | 0.72 | | | | | | 0.03 | 0.19 | 0.43 | | 0.21 |
> | GenAD [Zheng et al. 2024] | 0.28 || 0.49 || 0.78 | | 0.5* | | | | | | 0.08 | 0.14 | 0.34 | | 0.19 |
> | SparseDrive-S [Sun et al. 2024] | 0.29 || 0.58 || 0.96 | | 0.61 | | | | | | 0.01 | 0.05 | 0.18 | | 0.08 |
> | DriveTransformer-Large [Jia et al. 2025] | 0.16 || 0.30 || 0.55 | | **0.33** | | | | | | 0.01 | 0.06 | 0.15 | | 0.07 |
> | HiP-AD [Tang et al. 2025] | 0.28 || 0.53 || 0.87 | | 0.56 | | | | | | 0.01 | 0.05 | 0.15 | | 0.07 |
> | AlignDrive | 0.38 || 0.73 || 1.23 | | 0.78 | | | | | | 0.01 | 0.04 | 0.14 | | **0.06** |
>
> [1] Zhiqi Li, Zhiding Yu, Shiyi Lan, Jiahan Li, Jan Kautz, Tong Lu, and Jose M Alvarez. Is ego status
> all you need for open-loop end-to-end autonomous driving? In Proceedings of the IEEE/CVF
> Conference on Computer Vision and Pattern Recognition, pp. 14864–14873, 2024b.
>
> [2] Daniel Dauner, Marcel Hallgarten, Andreas Geiger, and Kashyap Chitta. Parting with misconceptions about learning-based vehicle motion planning. In Conference on Robot Learning, pp.
> 1268–1281. PMLR, 2023.
>
> [3] Jiang-Tian Zhai, Ze Feng, Jinhao Du, Yongqiang Mao, Jiang-Jiang Liu, Zichang Tan, Yifu Zhang, Xiaoqing Ye, and Jingdong Wang. Rethinking the open-loop evaluation of end-to-end autonomous
> driving in NuScenes. arXiv preprint arXiv:2305.10430, 2023.

---

> > ### Comment · Reviewer_GeYV · 2025-11-27
> >
> > Thanks for the addtional experiments and comparisons. I will maintain my score.

---

### Official Review · Reviewer_HWWw · 2025-10-31

**Soundness:** 3
**Presentation:** 3
**Contribution:** 3
**Rating:** 8
**Confidence:** 4

**Summary:**

This paper explore the planning representation problem in closed-loop end-to-end driving. By identifying the problem in previous parallel lateral and longitudinal prediction methods, this paper proposes a semi-coupled lateral and longitudinal planning framework, which first predicts driving path for lateral control, and the displacement along the path for longitudinal control. A planning-oriented augmentation is further proposed based on the planning framework.

**Strengths:**

* A semi-coupled planning framework is proposed, which first predicts driving path for lateral control, and use this path as the prior for longitudinal planning. Longitudinal planning is reformulated as a 1D prediction problem along the drive path, focusing on dynamic interactions.
* With the planning framework, a planning-oriented data augmentation is proposed to generate diverse safety-critical scenarios.
* The method achieves SOTA close-loop performance on challenging Bench2Drive benchmark.

**Weaknesses:**

* Though the data augmentation is useful, the agent encode module only use bounding box state and category, would it lose some appearance information extracted from image and contrary to the end-to-end philosophy?
* The longitudinal anchor set on candidate path is not described very clear, making it a little bit hard to understand.

**Questions:**

* As stated in introduction, the static scene elements are captured by drive path, why cross-attention with map queries is still needed in Contextual Interaction?
* In Agent Insertion, a virtual agent is randomly initialized with a randomly sampled state. First, is it reasonable that some cases will not follow road driving rules (in Fig.6., some vehicles do not drive along the road). Second, can the sample process be more efficient, making more threatening case than random sample?

---

> ### Author Response · Authors · 2025-11-21
>
> Thanks for your acknowledgement and kind advice. Regarding your concerns, we give responses below:
>
> > **Q1. Agent feature encoding**
>
> We agree that using only the bounding box state and category may omit certain low-level appearance cues. However, these structured attributes (e.g., position, velocity, and category) **already capture the most critical dynamic information that directly influences ego-vehicle trajectory prediction**. In practice, such representations are sufficient to model agent interactions in planning and have proven effective in supporting our data augmentation strategy.
> Regarding the end-to-end property, our framework remains differentiable: **the model takes raw images as input, and the losses of both the drive path and longitudinal planning branches are back-propagated to the image feature encoder**.
>
> > **Q2. Details of longitudinal anchor set**
>
> The longitudinal anchor set on each candidate path consists of a series of displacements under a constant-speed assumption. These anchors serve as reference displacements for the longitudinal planning, enabling structured and multimodal longitudinal planning along each path. We will clarify this description in the revised manuscript.
>
> > **Q3. Cross-attention with map queries**
>
> Although the drive path already captures certain static scene structures, cross-attention with map queries remains beneficial in Contextual Interaction.
> In longitudinal planning, **the model can benefit from map elements, such as stop lines and lane-ending cues, which provide important context for decisions like slowing down or yielding**. These elements are not fully represented by the drive path alone.
>
> > **Q4. Agent insertion strategy**
>
> Although some inserted agents do not strictly follow the road geometry, the augmentation remains meaningful because it is designed to **prioritize interactions with dynamic agents** rather than road alignment. This helps the model learn robust agent behaviors, which is the primary goal of our framework. Additionally, we chose the current random sampling strategy for its simplicity and ease of implementation, and it does not impose fundamental limitations on the framework. More sophisticated sampling strategies could indeed generate more diverse scenarios, which we plan to explore in future work.

---

### Author Response · Authors · 2025-11-21

We have included video samples in the **supplementary materials** to illustrate the capabilities of our AlignDrive across diverse driving scenarios. Please view the videos via the **index.html** file.

---

### Author Response · Authors · 2025-12-02
**Global response**

We sincerely thank all the reviewers for their thoughtful and constructive feedback, which has been both encouraging and insightful. We are pleased to see that the reviewers appreciate the following aspects of our work:

- A well-motivated and effectively designed framework that coordinates lateral and longitudinal planning and incorporates planning-oriented data augmentation(HWWw, GeYV, zjeJ, dQSN)

- The strong empirical performance, with significant improvements and new SOTA results.(HWWw, GeYV, zjeJ, dQSN)

- Comprehensive ablation studies and clear implementation details that validate the necessity of each component and support reproducibility. (GeYV, zjeJ)


We have carefully considered the reviewers' comments and revised our manuscript accordingly. The updated content is highlighted in magenta for clarity. Below is a brief outline of the revisions:

- Based on the suggestions of GeYV, zjeJ, and dQSN, we have conducted experiments on the real-world dataset nuScenes [1] in Sec. 4.3.

- In response to the suggestions of GeYV and zjeJ, we have refined Figures 1 and 2 and updated their captions to improve readability and clarity.

- At the suggestion of zjeJ, we have provided clearer explanations of the crucial concepts in our paper.

[1] Caesar, Holger, et al. "nuscenes: A multimodal dataset for autonomous driving." CVPR2020

---

### Meta-Review · Area_Chair_ZBee · 2026-01-10

**Summary:**

This paper proposes a novel end-to-end autonomous driving planning model called AlignDrive. Unlike previous approaches that jointly predict lateral and longitudinal displacements, this work first predicts the driving path, and then estimates the offset along this path. This strategy also effectively improves the planning performance.

**Reviewer Concerns:**

This work is reviewed by four expert reviewers with preliminary scores as 2, 8, 4, 6. The major concerns are summarized as follows:

- Discussion on the design choise of the agent encode module
- Technical details not explained in a clear way
- Lack of comparison with other notable previous works
- Other datasets extension, e.g. nuScenes and Navsim
- Incremental contribution
- Incomplete evaluation
- Writing clarity

**Reviewer Scores:**

Authors have done a good job addressing most of the concerns.

Three of the reviewers have responded in a timely manner. The major concerns regarding the technical contribution still remains. In particular, the

- novelty, AC felt in align with the reviewer that the idea of decoupling trejectory regression is not as distinct as previous works. Although authors have responded in great details.

- Incomplete experiments. Navsim results are not provided.

- Limitations of the open-loop L2 metric. This is not addressed well in the rebuttal. The collision-rate metric on nuScenes is already close to saturation. The L2 metric on nuScenes shows a clear gap compared with prior models.

AC read the paper, rebuttal and review comments. The manuscript would be greatly enhanced if authors revise the work based on review comments and submit it at future venue.

---

### Decision · Program_Chairs · 2026-01-26

Reject